# SubTrack++ : Gradient Subspace Tracking for Scalable LLM Training

**Sahar Rajabi, Nayeema Nonta, Sirisha Rambhatla**
Critical ML, Department of Management Science and Engineering, University of Waterloo
{srajabi, nnonta, srambhatla}@uwaterloo.ca

## Abstract

Training large language models (LLMs) is highly resource-intensive due to their massive number of parameters and the overhead of optimizer states. While recent work has aimed to reduce memory consumption, such efforts often entail trade-offs among memory efficiency, training time, and model performance. Yet, true democratization of LLMs requires simultaneous progress across all three dimensions. To this end, we propose SubTrack++ that leverages Grassmannian gradient subspace tracking combined with projection-aware optimizers, enabling Adam's internal statistics to adapt to subspace changes. Additionally, employing recovery scaling, a technique that restores information lost through low-rank projections, further enhances model performance. Our method demonstrates SOTA convergence by exploiting Grassmannian geometry, **reducing training wall-time by up to 65%** compared to the best performing baseline, LDAdam, while preserving the reduced memory footprint. Code is at `https://github.com/criticalml-uw/SubTrack`.

## 1 Introduction

LLMs have demonstrated state-of-the-art performance across a wide range of tasks and are rapidly growing in popularity. However, training and fine-tuning these models require significant resources, including extensive hardware and time, which limits their practicality for many applications and increases their environmental impact and carbon footprint. [Zhao et al., 2024, Jaiswal et al., 2024, Muhamed et al., 2024, Miles et al., 2024, Modoranu et al., 2024, Hao et al., 2024, Li et al., 2024].

Several techniques have been proposed to mitigate memory bottlenecks[Chen et al., 2016, Rajbhandari et al., 2020]. LoRA [Hu et al., 2021] and other low-rank adaptation methods [Dettmers et al., 2024, Hu et al., 2021, Yaras et al., 2024, Lialin et al., 2023, Renduchintala et al., 2024, Xia et al., 2024, Miles et al., 2024] have gained popularity by optimizing a reduced set of parameters. Such approaches often assume a low-rank parameter space, which can lead to suboptimal performance. In addition, methods like BAdam [Luo et al., 2024] and Block-LLM [Ramesh et al., 2024], utilize block coordinate descent to optimize parameter subsets, achieving memory savings at the cost of reduced accuracy.

However, memory requirements extend beyond trainable parameters, with a significant portion consumed by the optimizer's states [Zhao et al., 2024]. Recent efforts have focused on reducing this space while targeting full parameter training [Li et al., 2023, Anil et al., 2019, Lv et al., 2024, Dettmers et al., 2022, Zhang et al., 2024, Modoranu et al., 2024, Zhao et al., 2024, Muhamed et al., 2024]. Leveraging the low-dimensional nature of gradients during gradient descent [Gur-Ari et al., 2018, Schneider et al., 2024, Yaras et al., 2023] , GaLore [Zhao et al., 2024] reduces memory usage by projecting gradients into a low-rank subspace and periodically updating this approximation via singular value decomposition (SVD). While SVD offers optimal low-rank approximation [Robert et al., 2025], it can pose several challenges. First, it is computationally intensive, and alternatives which use random projections [Zhu et al., 2025] or approximation methods to estimate dominant singular values [Robert et al., 2025, Liang et al., 2024], match or outperform SVD in practice.

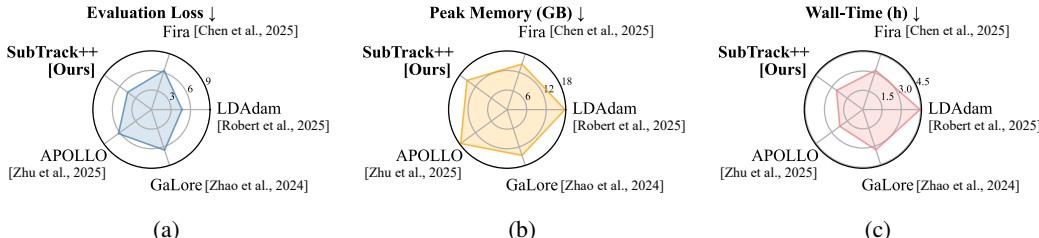

(a)  (b)  (c)

Figure 1: We compare baselines on pre-training a 1B-parameter model. (a) SubTrack++ achieves the lowest evaluation loss across all methods. (b) Its peak memory usage is significantly lower than APOLLO and LDAdam, and on par with GaLore and Fira. (c) In terms of wall-time, SubTrack++ incurs minimal overhead relative to APOLLO and is markedly faster than GaLore, Fira, and LDAdam. Overall, SubTrack++ outperforms all baselines in evaluation loss while matching or exceeding them in memory and runtime efficiency.

Moreover, SVD is sensitive to noise [Vaswani et al., 2018, He et al., 2025] and tends to degrade in late training stages when gradients are small, often hindering convergence [He et al., 2025].

Geometry-based methods have shown strong performance in various machine learning applications [Zhang et al., 2018, Balzano et al., 2011, He et al., 2011, Blocker et al., 2023]; Grassmannian is the manifold of all subspaces of dimensions $r$ in a space of dimensions $d$, and using Grassmannian for subspace tracking has led to structurally embedded information, lower computational complexity, and improved performance [Zhang et al., 2018, Balzano et al., 2011, He et al., 2011, Blocker et al., 2023, Chakraborty et al., 2017, Zhang and Balzano, 2016]. This line of work has demonstrated robustness and efficiency in high-dimensional, noisy environments [Zhang et al., 2018, Balzano et al., 2011, He et al., 2011, Balzano et al., 2018, Chakraborty et al., 2017]. Their natural robustness against perturbations and strong theoretical guarantees make them particularly well-suited to tracking the evolving gradient subspaces encountered in LLM training.

To this end, we employ subspace tracking on Grassmannian geodesics (Figure 2), to develop a geometry-based, time- and memory-efficient training method. This way, instead of reconstructing low-rank approximations via expensive SVD, we can efficiently leverage previously computed subspaces and the estimation error to better adjust the projection. We also incorporate subspace shifts into Adam's first and second momentum update rules, ensuring proper alignment with coordinate changes using a Projection-Aware Optimizer [Robert et al., 2025]. Additionally, we recover and scale (Recovery Scaling) the gradient information lost during low-rank projection by utilizing the scaling information of the low-rank, state-full optimizer Chen et al. [2025], Zhu et al. [2025].

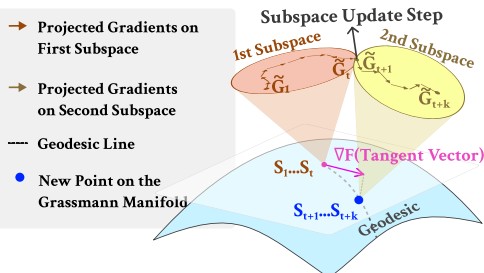

Figure 2: Visualization of Grassmannian subspace tracking: Between subspace updates, gradients are projected onto a fixed subspace. The tangent vector $\nabla F$ is computed via the derivative of a loss function, measuring the subspace estimation error. The subspace is then updated by moving along the corresponding geodesic, determined by $\nabla F$ to minimize estimation error.

To summarize, SubTrack++ is a projection-aware geometry-based approach that supports full-parameter training and incorporates recovery scaling, offering superior time efficiency compared to SVD or PowerSGD methods (e.g., GaLore [Zhao et al., 2024, Chen et al., 2025, Robert et al., 2025], while maintaining GaLore's memory footprint. It also outperforms online PCA subspace tracking methods [Liang et al., 2024] and achieves SOTA convergence and evaluation loss across all strong baselines; with comparison provided in Figure 1 and Figure 8.

## 2 SubTrack++

**Challenges of Low-Rank Optimization.** Projecting gradients into a low-rank subspace reduces memory footprint and enables scalable LLM training, but it introduces important trade-offs. First, gradient subspaces require adaptive tracking; while SVD-based methods can capture these shifts [Zhao et al., 2024, Chen et al., 2025], they are computationally expensive. To address this, recent work has explored cheaper approximations and random projections [Robert et al., 2025, Liang et al.,

2024, Zhu et al., 2025, He et al., 2025]. Furthermore, optimizers like Adam assume a fixed coordinate system, and subspace changes must be reflected in their internal states for a consistent momentum updates. Finally, low-rank projections inherently discard some gradient components, that recovering and utilizing these discarded signals can boost performance [Chen et al., 2025, Zhu et al., 2025].

**Overview.** SubTrack++ , a memory- and time-efficient method, embeds geometric insights into low-rank optimization, and improves efficiency and performance via three core components: 1) Grassmannian subspace tracking that refines projections via estimation error and subspace history; 2) projection-aware optimizer which adapts Adam's state to account for evolving subspaces; and 3) recovery scaling that restores lost information by scaling discarded gradient components.

**Subspace Tracking.** We frame subspace estimation as selecting a point on the Grassmannian, the manifold of all $d$-dimensional subspaces in an $n$-dimensional space [Bendokat et al., 2024]. This perspective offers three key benefits: 1) It refines the subspace using prior subspaces and estimation error, avoiding full reinitialization as done in Zhao et al. [2024], Chen et al. [2025], He et al. [2025]. 2) Updates rely on lightweight algebraic operations. 3) Controlled subspace shifts improve robustness against noise and abrupt changes. Similar to GaLore [Zhao et al., 2024], our proof in Theorem 3.2 shows that applying the projection to linear layers of LLMs preserves convergence while significantly reducing optimizer state memory. The initial subspace is computed using SVD as shown in (1). $G_0 \in \mathbb{R}^{m \times n}$ is the gradient matrix at step 0; $U$, $S$, and $V$ are its SVD components, and $r$ is the specified rank.

$$G_0 = USV^\top \approx \sum_{i=1}^{r} s_i u_i v_i^\top \tag{1}$$

At each step, gradients are projected onto the subspace of left singular vectors if $m \leq n$, and right singular vectors otherwise; optimizing memory usage [Zhao et al., 2024]. We assume $m \leq n$ without loss of generality, so the subspace is represented by $S_t \in \mathbb{R}^{m \times r}$ ($S_0 = [u_1, ..., u_r]$), an orthonormal basis spanning the top-$r$ directions. The gradient is projected as $\widetilde{G}_t = S_t^\top G_t \in \mathbb{R}^{r \times n}$, and the optimizer operates in this reduced space, significantly lowering memory and state overhead.

To account for subspace drift, the core subspace is updated every $k$ steps (the subspace update interval) by minimizing the cost function (2), which measures its Euclidean distance to the current gradient.

$$F(S_t) = \min_A \|S_t A - G_t\|_F^2, \tag{2}$$

where $A$ is the solution to the least squares problem. The derivative of (2) with respect to $S_t$ is given in (3), and the residual $R = G_t - S_t A$ lies in the orthogonal complement of $S_t$. To update the subspace, we calculate the tangent vector $\nabla F$ on the Grassmannian, as shown in (4) based on Edelman et al. [1998], where the second equality holds because $R$ is orthogonal to $S_t S_t^\top$.

$$\frac{\partial F}{\partial S_t} = 2(S_t A - G_t)A^\top = -2RA^\top \tag{3}$$

$$\nabla F = (I - S_t S_t^\top)\frac{\partial F}{\partial S_t} = \frac{\partial F}{\partial S_t} = -2RA^\top \approx \widehat{U}_F \widehat{\Sigma}_F \widehat{V}_F^\top \tag{4}$$

For optimizing the loss function (2), the subspace should be moved in the direction of $-\nabla F$ to reduce the estimation error. However, to control subspace changes, SubTrack++ computes a rank-1 approximation of $\nabla F$, determined by its largest singular value and the corresponding singular vector obtained from its SVD, represented as $\widehat{U}_F \widehat{\Sigma}_F \widehat{V}_F^\top$. This approximation is then used for subspace update. As shown by Edelman et al. [1998], Bendokat et al. [2024], we can move along a Grassmannian geodesic guided by rank-1 estimation of $-\nabla F$, with a step-size $\eta$, as presented in (5).

$$S_{t+1}(\eta) = (S_t \widehat{V}_F \quad \widehat{U}_F) \begin{pmatrix} \cos \widehat{\Sigma}_F \eta \\ -\sin \widehat{\Sigma}_F \eta \end{pmatrix} \widehat{V}_F^\top + S_t(I - \widehat{V}_F \widehat{V}_F^\top) \tag{5}$$

This update rule preserves the orthonormality of $S_{t+1}$, ensuring it remains on the Grassmannian. The last term in (5), projects the previous subspace onto the orthogonal complement of $\widehat{V}_F$, ensuring that the portion of $S_t$ which has not been updated in this step is still included.

**Projection-Aware Optimizer.** In Adam, the first and second momentum update rules are as shown in (6) and (7), respectively (reminder: $\widetilde{G}_t$ is the projection of gradient into low-rank subspace).

$$M_t \leftarrow \beta_1 \cdot M_{t-1} + (1 - \beta_1) \cdot \widetilde{G}_t \tag{6}$$

$$\mathcal{V}_t \leftarrow \beta_2 \cdot \mathcal{V}_{t-1} + (1 - \beta_2) \cdot \widetilde{G}_t^2 \tag{7}$$

**Algorithm 1** SubTrack++

( Subspace Tracking , Projection-Aware Optimizer , Recovery Scaling , Regular Adam )

---

**Require:** $W_t, G_t \in \mathbb{R}^{m \times n}$ with $m \leq n$ (w.l.o.g.), learning rate $\alpha$, decay rates $\beta_1$ and $\beta_2$, SubTrack++ step-size $\eta$, rank $r$, subspace update interval $k$, recovery scaling limiter factor $\zeta$. We use $\oslash$ to denote Hadamard division.

---

$S_0 \leftarrow U[:, :r]$, where $U, S, V \leftarrow \text{SVD}(G_0)$              {Initializing First Subspace}

**for** $t = 0, \ldots, T$ **do**

 **if** $t \bmod k == 0$ **then**

  $G_{lr} = \arg\min_A \|(S_{t-1}A - G_t)\|^2$, and $R = G_t - S_{t-1}G_{lr}$

  $\nabla F = -2RG_{lr}^\top \approx \widehat{U}_F \widehat{\Sigma}_F \widehat{V}_F^\top$

  $S_t = (S_{t-1}\widehat{V}_F \quad \widehat{U}_F) \begin{pmatrix} \cos \widehat{\Sigma}_F \eta \\ -\sin \widehat{\Sigma}_F \eta \end{pmatrix} \widehat{V}_F^\top + S_{t-1}(I - \widehat{V}_F \widehat{V}_F^\top)$

  $M_t \leftarrow \beta_1 \cdot (S_t^\top S_{t-1} M_{t-1}) + (1 - \beta_1) \cdot \widetilde{G}_t$    {$\widetilde{G}_t = S_t^\top G_t$: low-rank projection of $G_t$}

  $\mathcal{V}_t \leftarrow \beta_2 \cdot [(1 - \beta_2^{t-1})|(S_t^\top S_{t-1})^2 \cdot (\mathcal{V}_{t-1} - M_{t-1}^2) + (S_t^\top S_{t-1} \cdot M_{t-1})^2|] + (1 - \beta_2) \cdot \widetilde{G}_t^2$

 **else**

  $S_t = S_{t-1}$

  $M_t \leftarrow \beta_1 \cdot M_{t-1} + (1 - \beta_1) \cdot \widetilde{G}_t$

  $\mathcal{V}_t \leftarrow \beta_2 \cdot \mathcal{V}_{t-1} + (1 - \beta_2) \cdot \widetilde{G}_t^2$

 **end if**

 $\widetilde{G}_t^O = M_t \oslash \sqrt{\mathcal{V}_t + \epsilon}$, $\widehat{G}_t = S_t \widetilde{G}_t^O$    {$\widetilde{G}_t^O$: optimizer's output, $\widehat{G}_t$: projected-back gradients }

 $\phi_t(G_t)_i = \frac{\|\widetilde{G}_{t,:,i}^O\|}{\|\widetilde{G}_{t,:,i}\|}$, $\Lambda_t = \phi_t(G_t)(G_t - S_t \widetilde{G}_t)$    {We use $\oslash$ to denote Hadamard division.}

 **if** $\frac{\Lambda_t}{\Lambda_{t-1}} > \zeta$ **then** $\Lambda_t \leftarrow \frac{\Lambda_t}{\|\Lambda_t\|} \cdot \zeta \|\Lambda_{t-1}\|$

 $W_t \leftarrow W_{t-1} - \alpha \cdot \widehat{G}_t$ $-\alpha \cdot \Lambda_t$

**end for**

---

Inspired by methods such as LDAdam [Robert et al., 2025], we emphasize the importance of updating optimizer states in a projection-aware manner to account for shifting subspaces; otherwise, misaligned projections can distort the optimizer's performance. To address this, at each subspace update step, we modify Adam's original update rules in (6) and (7), replacing them with projection-aware counterparts represented in (8) and (9), which reflect subspace changes into optimizer statistics [Robert et al., 2025]. Further details regarding these projections can be found in Appendix C.

$$M_t \leftarrow \beta_1 \cdot (S_t^\top S_{t-1} M_{t-1}) + (1 - \beta_1) \cdot \widetilde{G}_t \tag{8}$$

$$\mathcal{V}_t \leftarrow \beta_2 \cdot [(1 - \beta_2^{t-1})|(S_t^\top S_{t-1})^2 \cdot (\mathcal{V}_{t-1} - M_{t-1}^2) + (S_t^\top S_{t-1} \cdot M_{t-1})^2|] + (1 - \beta_2) \cdot \widetilde{G}_t^2 \tag{9}$$

These projection-aware update rules enables Adam optimizer to track optimization dynamics precisely as the subspace evolves, achieving significantly better practical performance.

**Recovery Scaling.** Optimizer outputs $\widetilde{G}_t^O = M_t \oslash \sqrt{\mathcal{V}_t + \epsilon}$ ('$\oslash$' denotes Hadamard division), which is then projected back via $\widehat{G}_t = S_t \widetilde{G}_t^O$ to be used in weight update; however, low-rank projections inevitably discard some information in the full gradient matrix, that could enhance performance if properly utilized. Fira [Chen et al., 2025] observed that adaptive optimizers like Adam exhibit consistent scaling behaviour in low-rank and full-rank regimes. This suggests that the scaling information of the low-rank optimizer can be used to recover and rescale the discarded components of the gradient. A similar method is also employed in APOLLO [Zhu et al., 2025]. Consequently, an additional correction term is added to the standard weight update rule, as formalized in (10).

$$W_t \leftarrow W_{t-1} - \alpha \cdot \widehat{G}_t - \alpha \cdot \phi_t(G_t)(G_t - S_t \widetilde{G}_t) \tag{10}$$

Here $\alpha$ is the learning rate and $\phi_t(G_t)$ is the column-wise scaling factor computed based on the low-rank gradient representation, $\widetilde{G}_t$, and the optimizer's processed output, $\widetilde{G}_t^O$ as:

$$\phi_t(G_t)_i = \frac{\|\widetilde{G}_{t,:,i}^O\|}{\|\widetilde{G}_{t,:,i}\|} \tag{11}$$

Following Fira's observation [Chen et al., 2025], we employ a gradient-clipping-inspired mechanism to stabilize training. Specifically, we limit the growth rate of $\Lambda_t = \phi_t(G_t)(G_t - S_t \widehat{G}_t)$ by a factor $\zeta$, and apply a correction whenever it exceeds this threshold as:

$$\Lambda_t \leftarrow \frac{\Lambda_t}{\|\Lambda_t\|} \cdot \zeta \|\Lambda_{t-1}\| \tag{12}$$

By incorporating a geometry-aware perspective into low-rank optimization and applying this approach across all components of the training pipeline, SubTrack++ demonstrates robust and stable training, and achieves state-of-the-art performance while maintaining minimal memory footprint and wall-time. The overall flow of operations in SubTrack++ is illustrated in Algorithm 1.

## 3  Theoretical Analysis

In this section, we analyze the convergence of Grassmannian Subspace Tracking applied in Sub-Track++ using theoretical analysis. To begin, the general weights update rule is as follows:

$$W_t = W_0 - \alpha \cdot \sum_{t'=0}^{t'=t-1} \widehat{G}_{t'} \tag{13}$$

As previously mentioned, we use left projection if $m \leq n$, where $m$ and $n$ are the dimensions of the gradient matrix, and vice versa. Thus, $\widehat{G}_{t'}$ can be computed as shown in (14).

$$\widehat{G}_{t'} = \begin{cases} S_{t'} \rho_{t'}(S_{t'}^\top G_{t'}), & \text{if } m \leq n \\ \rho_{t'}(G_{t'} S_{t'}) S_{t'}^\top, & \text{otherwise} \end{cases} \tag{14}$$

Here, $S_{t'}$ is the projection matrix that projects the gradient onto the subspace, and $\rho_{t'}$ is representing the entry-wise regularizer used in the optimizer. If we use the full projection, then $\widehat{G}_{t'}$ will be computed as shown in (15); where $S_{t'}^l$ and $S_{t'}^r$ are the rank-$r$ left and right projection matrices.

$$\widehat{G}_{t'} = S_{t'}^l \rho_{t'}(S_{t'}^{l\top} G_{t'} S_{t'}^r) S_{t'}^{r\top} \tag{15}$$

**Definition 3.1 (L-continuity).** A function $f(X)$ has Lipschitz-continuity (L-continuity) if for any $X_1$ and $X_2$, $\|f(X_2) - f(X_1)\|_F \leq L\|X_2 - X_1\|_F$

**Theorem 3.2 (Convergence of Grassmannian Subspace Tracking).** *Suppose gradient has the following form with functions $A_i$, $B_i$, and $C_i$ being L-continuous as per **Def. 3.1** with constants $L_A$, $L_B$, and $L_C$ w.r.t. weight matrix $W_t$; and $\|W_t\|_F \leq M$; where $W_t$ denotes the weight matrix at step $t$, and $M$ is a scalar value,*

$$G = \sum_i A_i + \sum_i B_i W C_i.$$

*Now, define $\widehat{B}_{i,t} = (S_{i,t}^l)^\top B_i(W_t) S_{i,t}^l$ and $\widehat{C}_{i,t} = (S_{i,t}^r)^\top C_i(W_t) S_{i,t}^r$, where $S_{i,t}^l$ and $S_{i,t}^r$ are the rank-$r$ left and right projection matrices; $B_i(W)$ and $C_i(W)$ denote the dependence of $B_i$ and $C_i$ on the weight matrices $W_t$. Further letting $P_t = S_t^{l\top} G_t S_t^r$, and $\kappa_t = \frac{1}{N} \sum_i \lambda_{min}(\widehat{B}_{i,t}) \lambda_{min}(\widehat{C}_{i,t})$, where $\lambda_{min}(\cdot)$ denotes the minimum eigenvalue over each batch, and $N$ representing the number of samples in a batch. Assuming that the projection matrices remain constant during the training. Then for learning-rate $\mu$ and $min(\kappa_t) > (L_A + 2L_B L_C M^2)$, subspace tracking, with $\rho_t \equiv 1$ (the element-wise regularizer of the optimizer) satisfies:*

$$\|P_t\|_F \leq [1 - \mu(\kappa_{t-1} - L_A - 2L_B L_C M^2)]\|P_{t-1}\|_F.$$

*That is, $P_t \rightarrow 0$ and it converges.*

The proof of Theorem 3.2 is provided in Appendix A, based on Zhao et al. [2024]. While both GaLore and SubTrack++ assume the subspace remains unchanged for the proof of convergence, GaLore must limit these updates to ensure convergence, as each update can potentially change the entire subspace. In contrast, SubTrack++ leverages rank-1 updates to the subspace, preventing drastic changes with each update. While a deeper analysis of slowly changing subspaces and their impact on convergence remains an open problem, in practice, this allows SubTrack++ to perform more frequent updates.

Here we investigate the Grassmannian update rule presented in (5), which is a direct application of Grassmann geometry [Edelman et al., 1998, Bendokat et al., 2024].

Table 1: We compare evaluation loss (↓) for pre-training Llama-based architectures on the C4 dataset over 10k iterations. SubTrack++ outperforms all other baselines in nearly every configuration. The best results are marked in **bold**, with the second-best performance underlined. *LDAdam could not be run on the 7B configuration due to an out-of-memory error with our available resources.

| | 60M r=128 | 130M r=256 | 350M r=256 | 1B r=512 | 3B r=512 | 7B r=1024 |
|---|---|---|---|---|---|---|
| Full-Rank | 3.41 | 3.25 | 3.40 | 4.61 | 4.52 | 4.30 |
| GaLore [Zhao et al., 2024] | 4.02 | 3.61 | 3.62 | 6.53 | 6.57 | 5.55 |
| BAdam [Luo et al., 2024] | 7.86 | 7.08 | 7.62 | 7.28 | 7.12 | 6.76 |
| Online Subspace Descent [Liang et al., 2024] | 4.18 | 3.88 | 4.09 | 6.79 | 6.85 | 5.69 |
| LDAdam [Robert et al., 2025] | 3.52 | 3.44 | 3.67 | 4.70 | **4.39** | OOM* |
| Fira [Chen et al., 2025] | 3.80 | 3.55 | 3.56 | 6.31 | 6.50 | 6.83 |
| **SubTrack++** (Ours) | **3.43** | **3.24** | **3.29** | **4.52** | 4.50 | **4.63** |

**Definition 3.3 (Exponential Map).** The exponential map $\exp_p : T_p M \to M$ on a Riemannian manifold $M$ is a mapping that assigns the point $\gamma(1) \in M$ to each tangent vector $\Delta \in T_p M$, where $T_p M$ is the tangent space of $M$ at $p$, and $\gamma$ is the unique geodesic originating at $p$ with initial velocity $\Delta$. This map establishes a relationship between geodesics and the Riemannian exponential, such that $\gamma(t) = \exp_p(t\Delta)$ for $t \in \mathbb{R}$.

**Definition 3.4 (Stiefel Manifold).** The Stiefel manifold $St(n, p)$, parametrizes the set of all $n \times p$ orthonormal matrices $U$, each representing a rank-$p$ subspace of $\mathbb{R}^n$.

**Definition 3.5 (Grassmann Manifold).** The Grassmannian manifold $Gr(n, p)$ parametrizes the set of all $p$-dimensional subspaces of $\mathbb{R}^n$. Each point can be represented by a projection matrix $P = UU^\top$, where $U \in St(n, p)$.

**Theorem 3.6 (Grassmann Exponential).** *Let $P = UU^\top \in Gr(n, p)$ be a point on the Grassmannian, where $U \in St(n, p)$ is the orthonormal basis of the corresponding subspace. Consider a tangent vector $\Delta \in T_P Gr(n, p)$, and let $\Delta_U^{hor}$ denote the horizontal lift of $\Delta$ to the horizontal space at $U$ in the Stiefel manifold $St(n, p)$. Suppose the thin SVD of $\Delta_U^{hor}$ is given by $\Delta_U^{hor} = \hat{Q}\Sigma V^\top$, where $\hat{Q} \in St(n, r)$, $\Sigma = diag(\sigma_1, \ldots, \sigma_r)$ contains the nonzero singular values of $\Delta_U^{hor}$ with $r = \min(p, n - p)$, and $V \in St(p, r)$. The Grassmann exponential map, representing the geodesic emanating from $P$ in the direction $\Delta$, is given by:*

$$Exp_P^{Gr}(t\Delta) = [UV\cos(t\Sigma)V^\top + \hat{Q}\sin(t\Sigma)V^\top + UV_\perp V_\perp^\top],$$

*where $V_\perp \in \mathbb{R}^{p \times (p-r)}$ is any orthogonal complement of $V$.*

The proof of Theorem 3.6 can be found in Appendix B. Leveraging this theorem and our notation in section 2, one can easily verify that the subspace update rule is as follows:

$$S_{t+1}(\eta) = (S_t \widehat{V}_F \quad \widehat{U}_F) \begin{pmatrix} \cos \widehat{\Sigma}_F \eta \\ -\sin \widehat{\Sigma}_F \eta \end{pmatrix} \widehat{V}_F^\top + S_t(I - \widehat{V}_F \widehat{V}_F^\top)$$

This update rule generally converges to a stable subspace if the step size $\eta$ decreases over time [Balzano et al., 2011]. However, a decreasing step size can impair the ability to accurately track and adapt to subspace changes. Consequently, SubTrack++ uses a constant step size to effectively adjust subspaces. This approach does not hinder convergence, as proved in Theorem 3.2, which guarantees convergence as long as changes are controlled to maintain the stable subspace assumption.

## 4 Experiments and Results

We evaluated SubTrack++ across diverse models and datasets through pre-training and fine-tuning, measuring key metrics critical to LLM democratization.

**Pre-Training Experiments.** We pre-trained several Llama-based models on the C4 dataset, with results in Table 1. To ensure a fair comparison, we benchmarked against a diverse set of baselines.

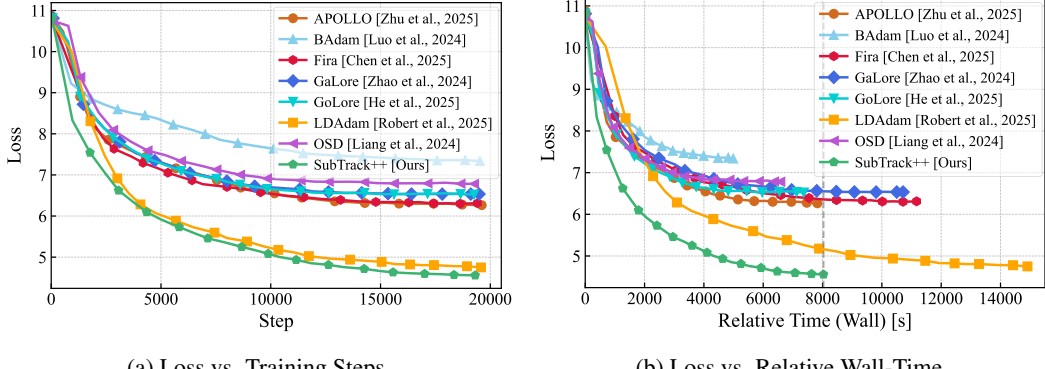

(a) Loss vs. Training Steps.

(b) Loss vs. Relative Wall-Time.

Figure 3: Comparison of baselines in pre-training Llama-1B architecture. (a) shows training loss (↓) versus training steps. (b) shows the same runs against wall-time. SubTrack++ outperforms all baselines; substantially reducing wall-time, especially compared to LDAdam, the top-performing baseline.

While all compared methods aim for memory-efficient training, their architectural principles yield distinct computational and convergence trade-offs. BAdam [Luo et al., 2024] achieves strong memory and time efficiency, yet its partial parameter update strategy compromises final performance. GaLore [Zhao et al., 2024] constrains gradients to a low-rank subspace estimated via periodic SVD, an approach sensitive to noise and costly to compute, while discarding information residing in the orthogonal complement. Fira [Chen et al., 2025] introduces norm-based recovery scaling to mitigate this information loss, but its reliance on frequent SVD still leads to substantial wall-time overhead. LDAdam [Robert et al., 2025] replaces SVD with PowerSGD-based iterative updates and a projection-aware optimizer that synchronizes internal states with evolving subspaces, improving convergence and stability but incurring high per-step costs due to continual subspace updates. It also adds an extended error-feedback mechanism to compensate for both gradient and optimizer compression to target GaLore's shortcomings. Online Subspace Descent (OSD) [Liang et al., 2024], our tracking-based baseline, further reduces complexity by employing Online PCA for subspace tracking.

SubTrack++ departs from these formulations by treating subspace evolution as a geometric tracking problem on the Grassmannian. It performs efficient rank-1 geodesic updates that reuse historical subspace information, inherently preserving consistency and avoiding the instability of discrete subspace resets. Simultaneously, it aligns Adam's internal states through projection-aware optimizer and leverages recovery scaling to reintegrate the lost information. This unified approach yields a new training paradigm that retains the memory efficiency of low-rank methods, while achieving on-par runtime efficiency compared to fastest baselines like APOLLO [Zhu et al., 2025]. SubTrack++ sets new state-of-the-art results across model scales, running 43% faster than LDAdam, the strongest prior baseline, on the 1B model, and 67% faster on the 3B model (see Table 6).

Figure 3 demonstrates the pre-training of a 1B-parameter Llama model across several baselines. SubTrack++ has the fastest convergence in both training steps and wall-time, highlighting the effectiveness of geometry-aware optimization in improving performance and reducing resource consumption. To further assess its generalization in longer training regimes and larger models, we extend training to 100k steps and compare SubTrack++ with GaLore. As shown in Figure 7, SubTrack++ converges substantially faster, achieving an evaluation loss of 3.37 (a significant improvement over the 10k-step setting), while GaLore reaches 4.64 under the same conditions.

As shown in Table 1, SubTrack++ occasionally outperforms full-rank training. This effect may be attributed to the implicit regularization introduced by low-rank projections, which can enhance generalization in overparameterized models. Similar trends have been reported in other studies [Robert et al., 2025, Zhu et al., 2025, Chen et al., 2025]. In addition, to further validate that convergence on the projected gradient reflects convergence of the original full-gradient, we measured both norms during Llama-1B pre-training. The full-gradient norm drops from $0.46 \rightarrow 0.08$, while the projected-gradient norm drops from $0.45 \rightarrow 0.05$, following nearly identical trajectories. This confirms that the optimization progress observed on the projected gradient accurately mirrors convergence in the original gradient space, consistent with prior low-rank gradient findings.

Hyperparameters of pre-training experiments are provided in Appendix E, with detailed runtime and memory reports in Appendix F.

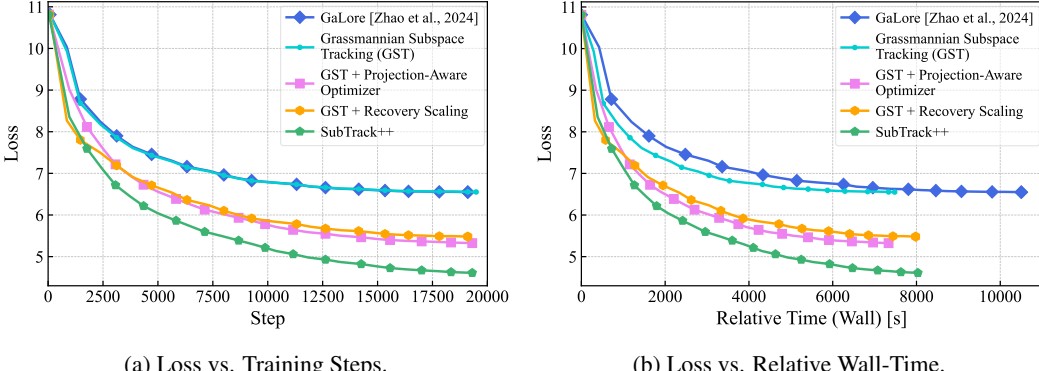

(a) Loss vs. Training Steps.

(b) Loss vs. Relative Wall-Time.

Figure 4: Ablation study comparing pure Grassmannian subspace tracking with incremental additions of the projection-aware optimizer and recovery scaling, leading to SubTrack++ . While Grassmannian tracking alone almost matches GaLore's step-wise convergence (a), it significantly reduces wall-time (b).

**Ablation Studies.** We conducted an ablation study to assess the individual and combined contributions of the projection-aware optimizer and recovery scaling, integrated with Grassmannian subspace tracking, on the 1B Llama model. Experimental settings are summarized in Table 4. As illustrated in Figure 4-b, Grassmannian subspace tracking alone substantially reduces wall-time compared to frequent SVD updates. Both the projection-aware optimizer and recovery scaling independently provide notable performance gains over baseline subspace tracking, lowering the loss from 6.53 to 5.43 and 5.28, respectively. Their combination, SubTrack++ , further improves the loss to 4.51, surpassing all baselines. Importantly, these improvements are achieved with only minimal increases in runtime and memory overhead, thanks to the efficiency of Grassmannian subspace tracking.

In addition, we conducted ablations on the subspace update rank and update frequency. As shown in Figure 5-a, more frequent updates can further improve performance, and the computational efficiency of SubTrack++ enables higher update frequencies with minimal overhead. However, excessively frequent updates may impede convergence. Figure 5-b shows that rank-1 updates achieve the best performance among all tested values. This suggests that making controlled, small adjustments to the underlying subspace helps maintain stability, while updating the subspace along the most informative direction prevents stagnation in a low-rank region and enhances generalization.

**Fine-Tuning Experiments.** RoBERTa-Base and RoBERTa-Large are fine-tuned on GLUE [Wang et al., 2019] and SuperGLUE [Sarlin et al., 2020] tasks; with the results presented in Table 7 and 8, respectively. We also conducted supervised fine-tuning of the Llama-2-7B-chat-hf model for one epoch on the Alpaca [Taori et al., 2023] dataset. In this experiment, SubTrack++ achieved 36% lower wall-time compared to GaLore and 65% compared to LDAdam. The results are presented in Table 9. More details and hyperparameters are provided in Appendix G.

**Time and Space Complexity.** Table 2 provides memory requirements of the optimizer states and the time complexity of the subspace update step considering an $m \times n$ gradient matrix with $m \leq n$.

GaLore [Zhao et al., 2024] and Fira [Chen et al., 2025] periodically perform SVD to estimate the underlying subspace, while LDAdam [Robert et al., 2025] relies on the faster PowerSGD to update the subspace at every iteration. In contrast, SubTrack++ employs Grassmannian-based subspace tracking at the same frequency as GaLore and Fira. Comparing the time complexities of these methods, highlights why SubTrack++ is significantly more efficient than SVD-based methods. A breakdown of the subspace update time complexity for SubTrack++ is shown in Appendix D. Additionally, the memory required for storing optimizer states in SubTrack++ , is equivalent to GaLore and other baselines.

Table 2: The optimizer's state parameter count and subspace update time complexity across baselines, given a gradient matrix of dimension $m \times n$ and projection rank $r$ where $r \ll m \leq n$. *LDAdam updates the subspace at every iteration, while other methods update it every $k$ steps.

|  | Optimizer Mem. | Subspace Update Time |
|---|---|---|
| Adam | $2mn$ | $-$ |
| LDAdam* | $mr + 2nr$ | $O(mnr)$ |
| GaLore, Fira | $mr + 2nr$ | $O(nm^2)$ |
| SubTrack++ | $mr + 2nr$ | $O(mnr)$ |

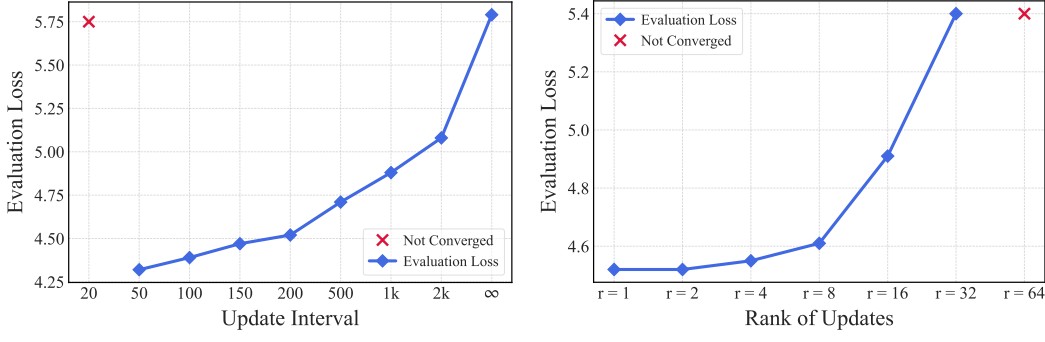

(a) Loss vs. Subspace Update Frequency.  (b) Loss vs. Subspace Update Rank.

Figure 5: Ablation results on (a) update frequency: decreasing the update interval (i.e., increasing the frequency) improves evaluation performance up to a point, but overly frequent updates hinder training convergence. (b) update rank: increasing the rank of updates degrades model performance, and beyond a certain threshold, can prevent convergence. These results emphasize the importance of controlled subspace adjustments.

**Robust Subspace Tracking.** Relying on SVD for subspace updates makes methods sensitive to noise and abrupt changes [He et al., 2025]. Figure 6 compares Grassmannian subspace tracking with GaLore's SVD on the Ackley function, highlighting how SVD causes erratic jumps, while our subspace tracking ensures robust optimization. GaLore struggles to reach the global minimum within 100 steps at scale factor 1, and although increasing the scale factor to 3 improves performance, it amplifies jumps that hinder convergence in non-convex settings, revealing sensitivity to hyperparameters, noise, and abrupt changes. To empirically measure the robustness of SubTrack++ in tracking evolving subspaces, we quantified subspace drift using the norm of the tangent vector $\nabla F$, which reflects the deviation between the projected gradient and the original gradient matrix. On Llama-350M pre-training, this norm rapidly decays from 0.06 to below 0.0002 within 5-7 subspace updates and remains near zero thereafter, indicating highly stable and well-aligned subspace tracking. This quantitative stability supports our qualitative observations in Figure 6 and aligns with prior findings in Grassmannian optimization.

## 5   Related Works

**Parameter-Efficient Training.** Several works aim to improve the efficiency of training LLMs, addressing a growing demand as their popularity rapidly increases. Popular LoRA [Hu et al., 2021] significantly reduces memory requirements for fine-tuning LLMs by leveraging two low-rank trainable low-rank matrices. Dettmers et al. [2024] employ quantization techniques and paged optimizers to further reduce memory usage. Additionally, Yaras et al. [2024] introduce Deep LoRA to address overfitting issues and reducing the need for precise tuning of the rank parameter. Several other works have also extended LoRA to enhance the efficiency of training and fine-tuning LLMs [Lialin et al., 2023, Renduchintala et al., 2024, Xia et al., 2024, Pan et al., 2024]. Miles et al. [2024] propose compressing intermediate activations and reconstructing them during backpropagation to enhance memory efficiency. Yen et al. [2025] propose adjustments to the LoRA factorization that promotes balanced training and correspondingly update the optimizer's internal states (i.e., first and second moments) to remain consistent under the change of basis. Additionally, Hao et al. [2024] demonstrate that full-parameter fine-tuning is feasible by random projections on the gradient matrix, showing that LoRA essentially performs a down-projection of the gradient. BAdam [Luo et al., 2024] leverages the block coordinate descent framework to reduce memory consumption while maintaining capabilities comparable to Adam.

**Gradient Low-Rank Projection.** Several approaches aim to reduce optimizer states, as optimizers like Adam [Kingma and Ba, 2017] account for a significant portion of memory footprint [Li et al., 2023, Anil et al., 2019, Lv et al., 2024, Dettmers et al., 2022]. MicroAdam [Modoranu et al., 2024] tackles this by compressing the gradient space and utilizing the compression error through feedback loops. Adam-mini [Zhang et al., 2024] partitions model into blocks, assigning a single learning rate to each block to preserve performance while saving memory. Gur-Ari et al. [2018], Schneider et al. [2024], Yaras et al. [2023] suggest that a substantial portion of gradients lies within a largely consistent subspace. GaLore [Zhao et al., 2024] leverages this fact to reduce the optimizer's memory

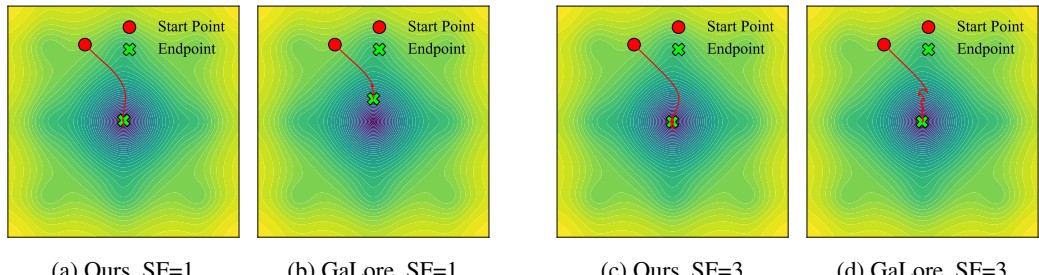

| (a) Ours, SF=1 | (b) GaLore, SF=1 | (c) Ours, SF=3 | (d) GaLore, SF=3 |

Figure 6: Comparison of Grassmannian subspace tracking (Ours) (a, c) and GaLore's SVD (b, d) on the Ackley Function over 100 optimization steps, with a subspace update interval of 10. SF stands for scale factor; with a scale factor of 1, GaLore fails to reach the global minimum due to abrupt jumps. At a scale factor of 3, while the minimum is reached, the jump length increases. This demonstrates SVD's sensitivity to noise and abrupt changes, highlighting the robustness of our subspace tracking method with its controlled subspace updates.

by projecting gradients into a low-rank subspace and then projecting them back for full parameter tuning. This approach has been integrated with other methods regarding efficient LLM training [Li et al., 2024]. However, not all layers' gradients evolve within a stable low-rank subspace. Jaiswal et al. [2024] identify layers where gradients evolve within a low-dimensional subspace and fine-tune only those layers, freezing the others to avoid inefficient low-rank updates. Grass [Muhamed et al., 2024] reduces memory usage by applying sparse projection matrices to the gradient. Ramesh et al. [2024] dynamically select and update a subset of parameters, for a fast and memory-efficient training. Fira [Chen et al., 2025] utilize a norm-based scaling method along with GaLore to maintain performance comparable to full-rank training. GoLore [He et al., 2025] addresses GaLore's convergence issues and employ random projection in latter steps as a solutions. LDAdam [Robert et al., 2025] performs optimization within lower-dimensional subspaces, incorporating a projection-aware optimization update rule and a generalized error feedback mechanism. Projection-Aware APOLLO [Zhu et al., 2025] approximates channel-wise learning rate scaling based on random projection. Also, Liang et al. [2024] introduce a dynamically evolving projection matrix updated via online PCA, enhancing the model's ability to navigate the parameter space efficiently without relying on expensive SVD.

**Geometric Subspace Updates.** A common approach in working with high-dimensional data is to project the data into a lower-dimensional space, and many studies focus on tracking these subspaces as they evolve. Balzano et al. [2011] introduce an incremental method for updating subspaces on the Grassmannian when the data is partially observed. Zhang and Balzano [2016] and Kasai [2017] propose methods to handle noise effect in tracking these subspaces. Furthermore, Blocker et al. [2023] present a method for evolving geodesic-based data in the Grassmannian for updating the subspace effectively. Mo et al. [2025] propose LORO, a low-rank pretraining method that performs Riemannian optimization on the manifold of fixed-rank matrices, enabling parameter- and memory-efficient training by updating low-rank factors via manifold-aware gradients and retractions.

# 6 Discussion and Conclusion

We propose SubTrack++ , a time- and memory-efficient approach that projects gradients into a low-rank subspace and uses Grassmannian subspace tracking to preserve the computed subspace while incorporating gradient components from the orthogonal complement. By integrating projection-aware optimizers that reflect subspace changes in Adam's internal statistics and utilizing the gradient information lost during low-rank projection, SubTrack++ achieves state-of-the-art convergence and accuracy across all baselines. While Grassmannian subspace tracking integrates seamlessly with various optimizers as a plug-and-play module, extending projection-aware optimization beyond the Adam family requires further design and investigation. Also benchmarking on models with more than 7B parameters was not feasible regarding our limited time and resources.

# Acknowledgments and Disclosure of Funding

Sirisha Rambhatla would like to acknowledge support of the Natural Sciences and Engineering Research Council of Canada (NSERC) Discovery Grant, RGPIN-2022-03512.

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

# A  Convergence of SubTrack++

**Theorem 3.2** (**Convergence of Grassmannian Subspace Tracking**). *Suppose gradient has the following form with functions $A_i$, $B_i$, and $C_i$ being L-continuous as per* **Def. 3.1** *with constants $L_A$, $L_B$, and $L_C$ w.r.t. weight matrix $W_t$; and $\|W_t\|_F \leq M$; where $W_t$ denotes the weight matrix at step $t$, and $M$ is a scalar value,*

$$G = \sum_i A_i + \sum_i B_i W C_i.$$

*Now, define $\widehat{B}_{i,t} = (S_{i,t}^l)^\top B_i(W_t) S_{i,t}^l$ and $\widehat{C}_{i,t} = (S_{i,t}^r)^\top C_i(W_t) S_{i,t}^r$, where $S_{i,t}^l$ and $S_{i,t}^r$ are the rank-r left and right projection matrices; $B_i(W_t)$ and $C_i(W_t)$ denote the dependence of $B_i$ and $C_i$ on the weight matrices $W_t$. Further letting $P_t = S_t^{l\top} G_t S_t^r$, and $\kappa_t = \frac{1}{N} \sum_i \lambda_{min}(\widehat{B}_{i,t}) \lambda_{min}(\widehat{C}_{i,t})$, where $\lambda_{min}(\cdot)$ denotes the minimum eigenvalue over each batch, and $N$ representing the number of samples in a batch. Assuming that the projection matrices remain constant during the training. Then for learning-rate $\mu$ and $min(\kappa_t) > (L_A + 2L_B L_C M^2)$, subspace tracking, with $\rho_t \equiv 1$ (the element-wise regularizer of the optimizer) satisfies:*

$$\|P_t\|_F \leq [1 - \mu(\kappa_{t-1} - L_A - 2L_B L_C M^2)]\|P_{t-1}\|_F.$$

*That is, $P_t \to 0$ and it converges.*

**proof.** To demonstrate that SubTrack++ converges to the global minimum during training, we begin by deriving the recursive form of the gradients.

Let $\otimes$ denote the Kronecker product. Then, $vec(AXB) = (B^\top \otimes A)vec(X)$.

By applying $vec$ to the gradient form given in the theorem, we obtain:

$$g_t = vec(G_t) = vec(\sum_i A_i + \sum_i B_i W C_i) = a_t - D_t w_t \tag{16}$$

where $g_t := vec(G_t)$, $w_t := vec(W_t)$, $a_t := \frac{1}{N} \sum_i vec(A_{i,t})$, and $D_t = \frac{1}{N} \sum_i C_{i,t} \otimes B_{i,t}$.

As defined in the theorem, let $P_t = S_t^{l\top} G_t S_t^r$. Its vectorized form can be expressed using the Kronecker product as follows:

$$
\begin{aligned}
p_t = vec(P_t) = vec(S_t^{l\top} G_t S_t^r) &= (S_t^{r\top} \otimes S_t^{l\top})vec(G_t) \\
&= (S_t^r \otimes S_t^l)^\top vec(G_t) = (S_t^r \otimes S_t^l)^\top g_t
\end{aligned}
\tag{17}
$$

Now recalling $\widehat{G}_t$ from (15), it can be written as:

$$\widehat{G}_t = S_t^l S_t^{l\top} G_t S_t^r S_t^{r\top}$$

Thus, its vectorized form will be:

$$
\begin{aligned}
vec(\widehat{G}_t) = \widehat{g}_t = vec(S_t^l S_t^{l\top} G_t S_t^r S_t^{r\top}) &= vec(S_t^l P_t S_t^{r\top}) \\
&= (S_t^r \otimes S_t^l)vec(P_t) = (S_t^r \otimes S_t^l)p_t
\end{aligned}
\tag{18}
$$

This is where the constant subspace assumption becomes necessary. To derive the recursive form of $g_t$, we assume that the projection matrices remain fixed throughout training, i.e., $S_t^r = S^r$ and $S_t^l = S^l$. Consequently, we can restate equations (17) and (18) as follows:

$$p_t = (S^r \otimes S^l)^\top g_t \tag{19}$$

$$\widehat{g}_t = (S^r \otimes S^l)p_t \tag{20}$$

Then we can write the recursive form of $g_t$:

$$
\begin{aligned}
g_t = a_t - D_t w_t &= (a_t - a_{t-1}) + (D_{t-1} - D_t)w_t + a_{t-1} - D_{t-1}w_t \\
&= e_t + a_{t-1} - D_{t-1}(w_{t-1} + \mu\widehat{g}_{t-1}) = e_t + g_{t-1} - \mu D_{t-1}\widehat{g}_{t-1}
\end{aligned}
\tag{21}
$$

where $e_t := (a_t - a_{t-1}) + (D_{t-1} - D_t)w_t$.

Note that in deriving (21), we utilized the general form of the weight update rule, $w_{t+1} = w_t - \mu g_t$, which can be rewritten as $w_t = w_{t+1} + \mu g_t$. By applying this rule along with (16), we arrive at the second equality in (21) as follows:

$$
\begin{aligned}
g_t = a_t - D_t w_t &= a_t - D_t w_t - g_{t-1} + g_{t-1} \\
&= a_t - D_t w_t - a_{t-1} + D_{t-1} w_{t-1} + a_{t-1} - D_{t-1} w_{t-1} \\
&= a_t - D_t w_t - a_{t-1} + D_{t-1}(w_t + \mu g_{t-1}) + a_{t-1} - D_{t-1}(w_t + \mu g_{t-1}) \\
&= a_t - D_t w_t - a_{t-1} + D_{t-1} w_t + \mu D_{t-1} g_{t-1} + a_{t-1} - D_{t-1} w_t - \mu D_{t-1} g_{t-1} \\
&= a_t - a_{t-1} + (D_{t-1} - D_t) w_t + a_{t-1} - D_{t-1}
\end{aligned}
$$

To obtain $p_t$ from this recursive formulation, we can left-multiply by $(S^r \otimes S^l)^\top$, as shown in (20):

$$
p_t = (S^r \otimes S^l)^\top e_t + (S^r \otimes S^l)^\top g_{t-1} - \mu(S^r \otimes S^l)^\top D_{t-1} \widehat{g}_{t-1} \tag{22}
$$

Now, based on (19) and (20), $p_t$ can be written as:

$$
p_t = (S^r \otimes S^l)^\top e_t + p_{t-1} - \mu(S^r \otimes S^l)^\top D_{t-1}(S^r \otimes S^l) p_{t-1} \tag{23}
$$

Let define:

$$
\begin{aligned}
\widehat{D}_t := (S^r \otimes S^l)^\top D_t (S^r \otimes S^l) &= \frac{1}{N} \sum_i (S^r \otimes S^l)^\top (C_{i,t} \otimes B_{i,t})(S^r \otimes S^l) \\
&= \frac{1}{N} \sum_i (S^{r\top} C_{i,t} S^r) \otimes (S^{l\top} B_{i,t} S^l)
\end{aligned} \tag{24}
$$

Then we can expand (23) and show that:

$$
p_t = (I - \mu \hat{D}_{t-1}) p_{t-1} + (S^r \otimes S^l)^\top e_t \tag{25}
$$

Note that $S^l$ and $S^r$ are orthonormal matrices. This is ensured because the subspace is initialized using the SVD of $G_0$, and the Grassmannian update rule provided in (5) preserves the orthonormality of the subspace matrices throughout training. Since $S^l$ and $S^r$ are orthonormal, we have $S^{l\top} S^l = I$ and $S^{r\top} S^r = I$. Consequently, we can bound the norm of the second term in (25) as follows:

$$
\|(S^r \otimes S^l)^\top e_t\|_2 = \|vec(S^{l\top} E_t S^r)\|_2 = \|S^{l\top} E_t S^r\|_F \leq \|E_t\|_F \tag{26}
$$

Here $E_t$ is the matrix form of $e_t$, and as declared before, $e_t := (a_t - a_{t-1}) + (D_{t-1} - D_t) w_t$, thus:

$$
E_t := \frac{1}{N} \sum_i (A_{i,t} - A_{i,t-1}) + \frac{1}{N} \sum_i (B_{i,t-1} W_t C_{i,t-1} - B_{i,t} W_t C_{i,t}) \tag{27}
$$

Next, we need to find an upper bound for the norm of each term in (27) to establish an upper bound for $\|E_t\|_F$. Based on the assumptions of the theorem, $A_i$, $B_i$, and $C_i$ exhibit L-Lipschitz continuity with constants $L_A$, $L_B$, and $L_C$, respectively. Additionally, $\|W_t\|_F$ is bounded by a scalar $M$. We have:

$$
\|A_t - A_{t-1}\|_F \leq L_A \|W_t - W_{t-1}\|_F = \mu L_A \|\tilde{G}_{t-1}\|_F \leq \mu L_A \|P_{t-1}\|_F \tag{28}
$$

In the first equality, we apply (13), while the last equality holds due to (20) and the orthonormality of the projection matrices. The subsequent two inequalities can be derived similarly using these equations.

$$
\begin{aligned}
\|(B_t - B_{t-1}) W_t C_{t-1}\|_F &\leq L_B \|W_t - W_{t-1}\|_F \|W_t\|_F \|C_{t-1}\|_F \\
&= \mu L_B L_C M^2 \|P_{t-1}\|_F
\end{aligned} \tag{29}
$$

$$
\begin{aligned}
\|B_t W_t (C_{t-1} - C_t)\|_F &\leq L_C \|B_t\|_F \|W_t\|_F \|W_{t-1} - W_t\|_F \\
&= \mu L_B L_C M^2 \|P_{t-1}\|_F
\end{aligned} \tag{30}
$$

We can now derive the bound for $\|E_t\|_F$ as follows:

$$
\begin{aligned}
\|E_t\|_F \leq \mu L_A \|\tilde{G}_{t-1}\|_F &\leq \mu L_A \|P_{t-1}\|_F + \mu L_B L_C M^2 \|P_{t-1}\|_F + \mu L_B L_C M^2 \|P_{t-1}\|_F \\
&= \mu(L_A + 2 L_B L_C M^2) \|P_{t-1}\|_F
\end{aligned} \tag{31}
$$

To calculate the norm bound for the first term in (25), we first need to establish the bounds for $\widehat{D}_t$. This involves estimating the minimum eigenvalue of $\widehat{D}_t$.

If we define $\gamma_{min,i,t} = \lambda_{min}(S^{l^\top}B_{i,t}S^l)\lambda_{min}(S^{r^\top}C_{i,t}S^r)$, then it follows that $\lambda_{min}((S^{l^\top}B_{i,t}S^l) \otimes (S^{r^\top}C_{i,t}S^r)) = \gamma_{min,i,t}$. Consequently, $\widehat{D}_t$ will satisfy the following inequality for every unit vector $\vec{v}$:

$$\vec{v}^\top \widehat{D}_t \vec{v} = \frac{1}{N}\sum_i \vec{v}^\top \left[(S^{l^\top}B_{i,t}S^l) \otimes (S^{r^\top}C_{i,t}S^r)\right]\vec{v} \geq \frac{1}{N}\sum_i \gamma_{min,i,t} \tag{32}$$

this actually provides a lower bound for eigenvalues of $\widehat{D}_t$, thus:

$$\lambda_{\max}(I - \mu\widehat{D}_{t-1}) \leq 1 - \frac{\mu}{N}\sum_i \gamma_{min,i,t-1} \tag{33}$$

considering the definition of $\kappa_t$ in the theorem, we can now easily show that:

$$\|P_t\|_F \leq [1 - \mu(\kappa_{t-1} - L_A - 2L_B L_C M^2)]\|P_{t-1}\|_F.$$

and completing the proof.

While SubTrack++ utilizes right/left projections to reduce memory consumption, the proof is presented using both projection matrices to ensure generality. Here, we demonstrate how the proof proceeds under the assumption $m \leq n$ (without loss of generality), which allows the use of the left projection matrix.

Using the left projection matrix, the current formulation of $P_t$, defined as $P_t = S_t^{l^\top}G_t S_t^r$, simplifies to $P_t = S_t^{l^\top}G_t$. Similarly, $\widehat{G}_t = S_t^l S_t^{l^\top}G_t S_t^r S_t^{r^\top}$ reduces to $\widehat{G}_t = S_t^l S_t^{l^\top}G_t$. From this point, the proof continues by substituting $S_t^r$ with the identity matrix, allowing the derivation of the vectorized forms of $g_t$, $\widehat{g}_t$, $p_t$, and related terms.

The remainder of the proof remains largely unaffected. It can be readily verified that the recursive formulation of $g_t$ is unchanged. Although the definition of $P_t$ is modified, it continues to satisfy the bounds required for convergence, ensuring that $P_t$ converges to 0 when the left projection matrix is used.

## B   Grassmann Exponential

**Theorem 3.6 (Grassmann Exponential).** *Let $P = UU^\top \in Gr(n,p)$ be a point on the Grassmannian, where $U \in St(n,p)$ is the orthonormal basis of the corresponding subspace. Consider a tangent vector $\Delta \in T_P Gr(n,p)$, and let $\Delta_U^{hor}$ denote the horizontal lift of $\Delta$ to the horizontal space at $U$ in the Stiefel manifold $St(n,p)$. Suppose the thin SVD of $\Delta_U^{hor}$ is given by $\Delta_U^{hor} = \hat{Q}\Sigma V^\top$, where $\hat{Q} \in St(n,r)$, $\Sigma = diag(\sigma_1,\ldots,\sigma_r)$ contains the nonzero singular values of $\Delta_U^{hor}$ with $r = \min(p,n-p)$, and $V \in St(p,r)$. The Grassmann exponential map, representing the geodesic emanating from $P$ in the direction $\Delta$, is given by:*

$$Exp_P^{Gr}(t\Delta) = [UV\cos(t\Sigma)V^\top + \hat{Q}\sin(t\Sigma)V^\top + UV_\perp V_\perp^\top],$$

*where $V_\perp \in \mathbb{R}^{p\times(p-r)}$ is any orthogonal complement of $V$.*

**proof.** Using Grassmannina mathematics, we know that every $\Delta \in T_P Gr(n,p)$ is of the form

$$\Delta = Q\begin{pmatrix} 0 & B^\top \\ B & 0 \end{pmatrix}Q^\top = \left[Q\begin{pmatrix} 0 & -B^\top \\ B & 0 \end{pmatrix}Q^\top, P\right] \tag{34}$$

Then the lift of $\Delta \in T_P Gr(n,p)$ to $Q = (U \quad U_\perp)$ can also be calculated explicitly as follows:

$$\Delta_Q^{hor} = [\Delta, P]Q = Q\begin{pmatrix} 0 & -B^\top \\ B & 0 \end{pmatrix} \tag{35}$$

To resume our proof, we need to define the orthogonal group and specifying its tangent space.

**Definition B.1** (**Orthogonal Group**). The orthogonal group $O(n)$ is defined as the set of all $n \times n$ matrices $Q$ over $\mathbb{R}$ such that $Q^\top Q = Q Q^\top = I_n$, where $Q^\top$ is the transpose of $Q$ and $I_n$ is the $n \times n$ identity matrix:

$$O(n) = \{Q \in \mathbb{R}^{n \times n} \mid Q^\top Q = I_n = Q Q^\top\}.$$

Then the tangent space of the orthogonal group $O(n)$ at a point $Q$, denoted $T_Q O(n)$, is defined as the set of matrices of the form $Q\Omega$, where $\Omega \in \mathbb{R}^{n \times n}$ is a skew-symmetric matrix, i.e., $\Omega^\top = -\Omega$:

$$T_Q O(n) = \{Q\Omega \mid \Omega \in \mathbb{R}^{n \times n}, \Omega^\top = -\Omega\}.$$

The geodesic from $Q \in O(n)$ in direction $Q\Omega \in T_Q O(n)$ is calculated via

$$\mathrm{Exp}_Q^O(tQ\Omega) = Q \exp_m(t\Omega), \tag{36}$$

If $P \in Gr(n,p)$ and $\Delta \in T_P Gr(n,p)$ with $\Delta_Q^{\text{hor}} = Q \begin{pmatrix} 0 & -B^\top \\ B & 0 \end{pmatrix}$, the geodesic in the Grassmannian is therefore

$$\mathrm{Exp}_P^{Gr}(t\Delta) = \pi^{OG}\left(Q \exp_m\left(t\begin{pmatrix} 0 & -B^\top \\ B & 0 \end{pmatrix}\right)\right). \tag{37}$$

where $\pi^{OG}$ is the projection from $O(n)$ to $Gr(n,p)$. If the thin SVD of $B$ is given by

$$B = U_\perp^\top \Delta_U^{\text{hor}} = U_\perp^\top \hat{Q} \Sigma V^\top$$

with $W := U_\perp^\top \hat{Q} \in St(n-p,r), \Sigma \in \mathbb{R}^{r \times r}, V \in St(p,r)$. Let $W_\perp, V_\perp$ be suitable orthogonal completions. Then,

$$\exp_m\begin{pmatrix} 0 & -B^\top \\ B & 0 \end{pmatrix} = \begin{pmatrix} V & V_\perp & 0 & 0 \\ 0 & 0 & W & W_\perp \end{pmatrix} \begin{pmatrix} \cos(\Sigma) & 0 & -\sin(\Sigma) & 0 \\ 0 & I_{p-r} & 0 & 0 \\ \sin(\Sigma) & 0 & \cos(\Sigma) & 0 \\ 0 & 0 & 0 & I_{n-p-r} \end{pmatrix} \begin{pmatrix} V^\top & 0 \\ V_\perp^\top & 0 \\ 0 & W^\top \\ 0 & W_\perp^\top \end{pmatrix},$$

which leads to the desired result when inserted into (37). For more mathematical details, you can refer to Edelman et al. [1998], Bendokat et al. [2024], or other useful resources on Grassmann geometry.

## C  Projection-Aware Optimizer

When projecting into a lower-dimensional space and tracking coordinate changes, we typically use orthonormal projection matrices to represent the subspaces and their multiplications to effect a change of basis. While this works well for purely linear operations, Adam's updates also incorporate non-linear elements.

Suppose the subspace changes at step $t$, transitioning from the subspace spanned by the orthonormal matrix $S_{t-1}$ to that spanned by $S_t$. Since both matrices are orthonormal—preserved by the Grassmannian update rule in (5)—the matrix $S_t^\top S_{t-1}$ represents the change of basis between the two subspaces. In other words, if $\mathbb{E}_{t-1} = (e_{t-1}^1, \ldots, e_{t-1}^r)$ and $\mathbb{E}_t = (e_t^1, \ldots, e_t^r)$ are orthonormal bases for the subspaces at steps $t-1$ and $t$, respectively, then the $i$th column of matrix $X$ transforms under the change of basis as $X_t^i = \sum_{j=1}^r \langle e_t^i, e_{t-1}^j \rangle X_{t-1}^j$, where $X_{t-1}^j$ is the $j$th column of $X$ based on the basis of the time step $t-1$.

$\mathbb{E}_{t,\beta}[.]$ denotes the exponential time-weighted expectation at time $t$ with decay rate $\beta$. Following the reinterpretation by Robert et al. [2025], Adam's first and second moment estimates can be expressed as $\widetilde{M}_t = \mathbb{E}_{t,\beta_1}\left[\widetilde{G}_t\right]$ and $\widetilde{\mathcal{V}}_t = \mathbb{E}_{t,\beta_2}\left[(\widetilde{G}_t)^2\right]$, where $\widetilde{G}_t$ denotes the low-rank representation of the gradient at time step $t$. As shown in (38), the first moment estimate can be transformed under a change of basis using the change-of-basis matrix, $S_t^\top S_{t-1}$. Notably, $\langle \widetilde{G}_t, e_t^i \rangle$ gives the $i$th column of $\widetilde{G}_t$ when the subspace has the basis $\mathbb{E}_t$. We use the superscripts to indicate a column of a matrix.

$$\mathbb{E}_{t,\beta_1}\left[\langle \widetilde{G}_t, e_t^i \rangle\right] = \sum_{j=1}^r \langle e_t^i, e_{t-1}^j \rangle \mathbb{E}_{t,\beta_1}\left[\langle \widetilde{G}_t, e_{t-1}^j \rangle\right] = \sum_{j=1}^r \langle e_t^i, e_{t-1}^j \rangle \widetilde{M}_t^j = \left(S_t^\top S_{t-1} \widetilde{M}_t\right)^i \tag{38}$$

Following the same approach, we can change the basis for the second moment estimate as described in (39).

$$
\mathbf{E}_{t,\beta_2}\left[\left(\langle \widetilde{G}_t, e_t^i\rangle\right)^2\right] = \sum_{j=1}^{r}\langle e_t^i, e_{t-1}^j\rangle^2\,\mathbf{E}_{t,\beta_2}\left[\left(\langle \widetilde{G}_t, e_{t-1}^j\rangle\right)^2\right]
$$
$$
+ \sum_{k\neq l}^{r}\langle e_t^i, e_{t-1}^k\rangle\langle e_t^i, e_{t-1}^l\rangle\,\mathbf{E}_{t,\beta_2}\left[\langle \widetilde{G}_t, e_{t-1}^k\rangle\langle \widetilde{G}_t, e_{t-1}^l\rangle\right]
$$
$$
= \sum_{j=1}^{r}\langle e_t^i, e_{t-1}^j\rangle^2\widetilde{\mathcal{V}}_t^j
$$
$$
+ \sum_{k\neq l}^{r}\langle e_t^i, e_{t-1}^k\rangle\langle e_t^i, e_{t-1}^l\rangle\widetilde{M}_t^k\widetilde{M}_t^l. \tag{39}
$$

In transitioning from the first equality to the second, we assume independence among the gradient coordinates. This enables us to approximate the covariance using a product of first-order moment estimates. This assumption is often reasonable in practice because we compute the SVD of the gradient and maintain an orthonormal subspace projection matrix, updating it along the Grassmannian geodesic to track the optimal subspace. Since SVD tends to diagonalize the covariance, the off-diagonal entries are typically negligible. Additionally, we clip any negative values to zero to ensure valid (non-negative) variance estimates. Moreover, to rewrite the second term in the final equality of (39), we employ the following equation:

$$
\sum_{k}\sum_{l}\langle e_t^i, e_{t-1}^k\rangle\langle e_t^i, e_{t-1}^l\rangle\widetilde{M}_t^k\widetilde{M}_t^l
$$
$$
= \sum_{k}\langle e_t^i, e_{t-1}^k\rangle^2\left(\widetilde{M}_t^k\right)^2 + \sum_{k\neq l}\langle e_t^i, e_{t-1}^k\rangle\langle e_t^i, e_{t-1}^l\rangle\widetilde{M}_t^k\widetilde{M}_t^l \tag{40}
$$

Given these, we can rewrite (39) as follows:

$$
\mathbf{E}_{t,\beta_2}\left[\left\langle \widetilde{G}_t, e_t^i\right\rangle^2\right] = \sum_{j}\langle e_t^i, e_{t-1}^j\rangle^2\widetilde{\mathcal{V}}_t^j +
$$
$$
\left[\sum_{k}\sum_{l}\langle e_t^i, e_{t-1}^k\rangle\langle e_t^i, e_{t-1}^l\rangle\widetilde{M}_t^k\widetilde{M}_t^l - \sum_{k}\langle e_t^i, e_{t-1}^k\rangle^2\left(\widetilde{M}_t^k\right)^2\right]
$$
$$
= \sum_{j}\langle e_t^i, e_{t-1}^j\rangle^2\left[\widetilde{\mathcal{V}}_t^j - \left(\widetilde{M}_t^j\right)^2\right] + \left(\langle e_t^i, e_{t-1}^j\rangle\widetilde{M}_t\right)^2
$$
$$
= \left(\left(S_t^\top S_{t-1}\right)^2\left[\widetilde{\mathcal{V}}_t - \widetilde{M}_t^2\right]\right)^i + \left(\left(S_t^\top S_{t-1}\widetilde{M}_t\right)^2\right)^i \tag{41}
$$

By applying (38) and (41), we can directly derive the update rules for the projection-aware optimizer, as expressed in (8) and (9).

## D   Time Complexity Analysis

Table 3 presents the time complexity breakdown for the subspace update step in the SubTrack++ algorithm assuming a $m \times n$ gradient matrix and rank $r$ projection matrix, where $r \ll m \le n$. As outlined in Algorithm 1, the subspace update step begins by solving the least squares problem (2) to estimate the optimal update for $S_t$, the $m \times r$ orthonormal matrix. This operation has a time complexity of $O(mr^2)$. Computing the residual and the partial derivative with respect to $S_t$ requires $O(mrn)$ and $O(mnr)$ time respectively. This is because the solution to the least squares problem, $A$, has shape $r \times n$ which is multiplied by $S_t$ in the residual $R = G_t - S_t A$, resulting in time complexity $O(mrn)$. The following operation for the partial derivative is $-2RA^T$, where the matrix multiplication has $O(mnr)$ complexity. The tangent vector computation (4) which involves an identity transformation and matrix multiplication has time complexity of $O(m^2 r)$.

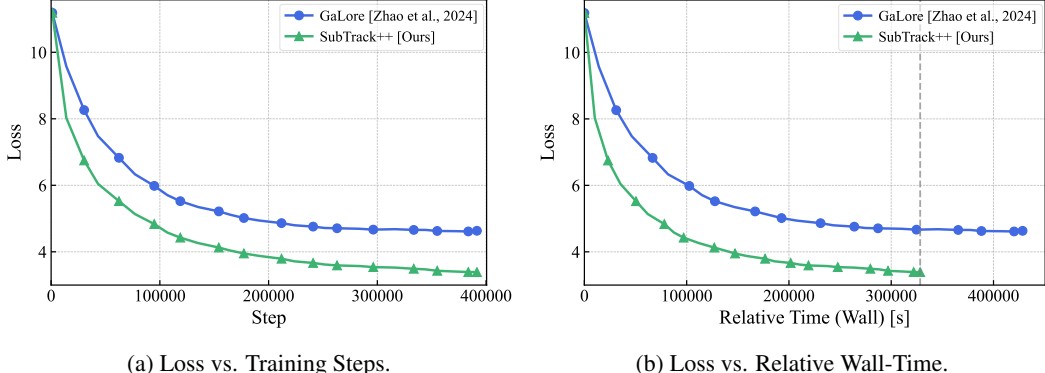

(a) Loss vs. Training Steps.                    (b) Loss vs. Relative Wall-Time.

Figure 7: Comparison of pre-training Llama-7B architecture for 100k iterations. (a) shows training loss (↓) versus training steps. (b) shows the same runs against wall-time. SubTrack++ outperforms GaLore; substantially reducing wall-time.

Table 3: Time Complexity for SubTrack++ Subspace Update

The rank-1 approximation step uses largest singular value from the SVD of the $m \times r$ tangent vector, and has time complexity of $O(mr^2)$. Finally, the update rule as shown in (13) which has a time complexity of $O(mr^2)$. The overall complexity of the algorithm is dominated by the matrix multiplication calculations of time complexity $O(mnr)$. However, unlike GaLore, since we avoid computing SVD operation on the $m \times n$ gradient matrix, which has complexity of $O(nm^2)$, the overall update step in SubTrack++ is still more efficient with respect to time complexity.

| Computation Step | Time |
|---|---|
| Cost function | $O(mr^2)$ |
| Residual | $O(mrn)$ |
| Partial derivative | $O(mnr)$ |
| Tangent vector $\Delta F$ | $O(m^2r)$ |
| Rank-1 approximation of $\Delta F$ | $O(mr^2)$ |
| Update rule | $O(mr^2)$ |
| **Overall** | $O(mnr)$ |

# E   Pre-Training Llama-Based Architectures

We pre-trained all six Llama-based architectures for 10k iterations using hyperparameters reported in Table 4. To demonstrate the generalizability of the proposed method, we also present results from pre-training the 7B architecture with both SubTrack++ and GaLore for 100k iterations, as shown in Figure 7. SubTrack++ maintains its advantage in terms of faster convergence and superior performance. The hyperparameters of this run are identical to those reported in Table 4, except for the number of iterations which is 100k. Also the bar-chart version of Figure 1 is represented in Figure 8.

# F   Memory and Time Comparison

Table 5 presents the the peak memory consumption measured to compare SubTrack++ and other baselines. It shows that SubTrack++ requires on-par or better memory compared to GaLore [Zhao et al., 2024] while getting state-of-the-art results. As detailed in Table 2, all baselines except BAdam [Luo et al., 2024] use the same number of optimizer parameters; therefore, any differences in their peak memory consumption stem from variations in their runtime parameters.

Additionally, Table 6 presents the wall-time consumed by each baseline across all model architectures during pre-training. Each run is configured to include exactly 10 subspace updates for SubTrack++ and other baselines employing periodic subspace updates. Specifically, models ranging from 60M to 3B use a subspace update interval of 200, resulting in 2,000 total iterations, while the 7B models use an interval of 500, yielding 5,000 iterations. Experiments for the 60M to 3B models are conducted on an NVIDIA A100 GPU, while the 7B model experiments are run on an NVIDIA RTX A6000.

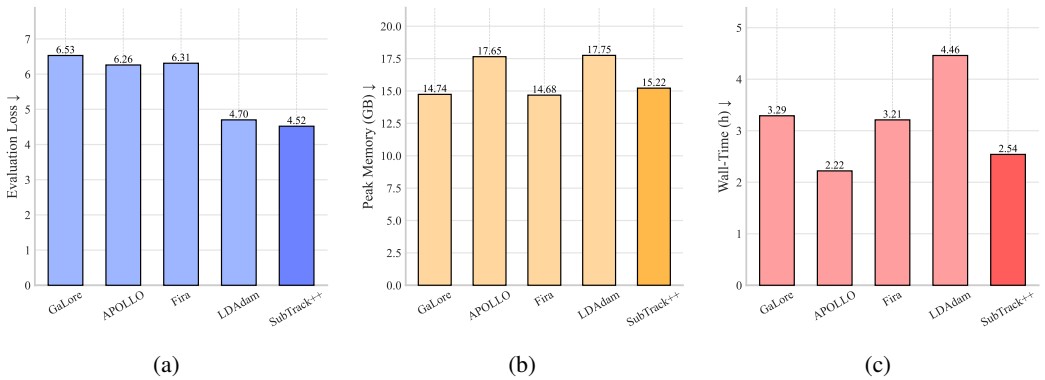

(a)                 (b)                 (c)

Figure 8: We compare baselines on pre-training a 1B-parameter model. (a) SubTrack++ achieves the lowest evaluation loss across all methods. (b) Its peak memory usage is significantly lower than APOLLO and LDAdam, and on par with GaLore and Fira. (c) In terms of wall-time, SubTrack++ incurs minimal overhead relative to APOLLO and is markedly faster than GaLore, Fira, and LDAdam. Overall, SubTrack++ outperforms all baselines in evaluation loss while matching or exceeding them in memory and runtime efficiency.

Table 4: Hyperparameters of pre-training Llama-based architectures.

|  |  | 60M | 130M | 350M | 1B | 3B | 7B |
|---|---|---|---|---|---|---|---|
| Architectural Parameters | Hidden | 512 | 768 | 1024 | 2048 | 2560 | 4096 |
|  | Intermediate | 1376 | 2048 | 2736 | 5461 | 6848 | 11008 |
|  | Heads | 8 | 12 | 16 | 24 | 32 | 32 |
|  | Layers | 8 | 12 | 24 | 32 | 32 | 32 |
| Shared Parameters | Learning Rate | 1e-3 | 1e-3 | 1e-3 | 1e-4 | 1e-4 | 1e-4 |
|  | Batch Size | 128 | 128 | 64 | 8 | 8 | 4 |
|  | Gradient Accumulation | 2 | 2 | 2 | 2 | 2 | 4 |
|  | Iterations | | | | 10k | | |
|  | Gradient Clipping | | | | 1.0 | | |
|  | Warmup Steps | | | | 1000 | | |
|  | scale | | | | 0.25 | | |
|  | dtype | | | | bfloat16 | | |
| Low-Rank Optimizer Methods Parameters | Rank | 128 | 256 | 256 | 512 | 512 | 1024 |
|  | Subspace Update Interval | 200 | 200 | 200 | 200 | 200 | 500 |
|  | SubTrack++ Step-Size | | | | 10000 | | |
| BAdam Parameters | Block Swithch Interval | | | | 100 | | |
|  | Switch Mode | | | | Random | | |

# G   Fine-Tuning Experiments

As described, we examined SubTrack++ on fine-tuning RoBERTa-base and RoBERTa-large models to evaluate them on GLUE and SuperGLUE benchmarks. The results for GLUE task is summarized in Table 7, and SuperGLUE in 8.

The hyperparameters for fine-tuning RoBERTa-base are detailed in Table 10, matching those reported in the GaLore [Zhao et al., 2024] for rank-8 subspaces, with a subspace update interval set at 500 iterations. We also fine-tuned RoBERTa-Large on SuperGLUE tasks using the hyperparameters from Luo et al. [2024], as detailed in Table 11, with the exception that we fine-tuned each task for 30 epochs.

The results of supervised fine-tuning of the Llama-2-7B-chat-hf model on the Alpaca dataset on an Nvidia-H100 GPU are presented in Table 9. The fine-tuning was performed for one epoch, and the corresponding hyperparameters are listed in Table 12.

Table 5: Peak memory consumption of pre-training Llama-based architectures on C4 dataset. The 7B models are trained using the 8-bit Adam optimizer, except for the runs marked with *. SubTrack++ demonstrates better or on-par memory compared to other low-rank methods that allows full-parameter training.

| | 60M r=128 | 130M r=256 | 350M r=256 | 1B r=512 | 3B r=512 | 7B r=1024 |
|---|---|---|---|---|---|---|
| Full-Rank | 16.86 | 25.32 | 28.67 | 18.83 | 34.92 | 50.50 |
| BAdam [Luo et al., 2024] | 13.34 | 20.01 | 16.45 | 9.18 | 14.75 | 22.35 |
| GaLore [Zhao et al., 2024] | 16.89 | 25.52 | 27.85 | 14.74 | 26.03 | 36.00 |
| Online Subspace Descent Liang et al. [2024] | 16.61 | 25.86 | 28.76 | 18.45 | 32.57 | 33.10 |
| LDAdam [Robert et al., 2025] | 17.18 | 25.94 | 28.03 | 18.45 | 32.57 | OOM* |
| Fira [Chen et al., 2025] | 16.39 | 24.99 | 27.33 | 14.68 | 25.62 | 47.84* |
| **SubTrack++** (Ours) | 16.40 | 25.06 | 27.42 | 15.22 | 25.54 | 49.82* |

Table 6: Wall-time comparison of pre-training Llama-based architectures on the C4 dataset. The number of iterations is set to ensure 10 subspace updates for methods using periodic subspace adjustments. SubTrack++ achieves the lowest wall-time among all baselines that support full-parameter training on large models. The 7B models are trained using the 8-bit Adam optimizer, except for the runs marked with *. Since this can impact wall-time comparisons, it is more appropriate to compare runs within the same cluster.

| | 60M r=128 | 130M r=256 | 350M r=256 | 1B r=512 | 3B r=512 | 7B r=1024 |
|---|---|---|---|---|---|---|
| Full-Rank | 524.0 | 1035.1 | 1396.4 | 974.9 | 1055.9 | 12726.2 |
| BAdam [Luo et al., 2024] | 511.3 | 779.2 | 961.6 | 798.6 | 1004.1 | 7283.7 |
| GaLore [Zhao et al., 2024] | 547.8 | 1094.2 | 1589.0 | 1729.5 | 2715.5 | 21590.4 |
| Online Subspace Descent [Liang et al., 2024] | 662.8 | 1228.2 | 1818.6 | 1438.7 | 1676.9 | 18221.9 |
| LDAdam [Robert et al., 2025] | 639.9 | 1342.2 | 2083.4 | 2780.9 | 4625.4 | OOM* |
| Fira [Chen et al., 2025] | 635.3 | 1180.8 | 1729.7 | 1938.5 | 2898.4 | 22554.3* |
| **SubTrack++** (Ours) | 627.6 | 1140.6 | 1593.2 | 1304.3 | 1517.6 | 16491.7* |

Table 7: Evaluating the performance of SubTrack++ and other baselines when fine-tuning RoBERTa-Base on GLUE tasks for $r = 8$. The performance is measured via Accuracy ($\uparrow$) for SST-2 and RTE tasks, F1 ($\uparrow$) for MRPC, Pearson Correlation ($\uparrow$) for STS-B, and Matthews Correlation ($\uparrow$) for COLA. The best results are marked in **bold**, with the second-best performance underlined.

| | COLA | STS-B | MRPC | RTE | SST-2 |
|---|---|---|---|---|---|
| Full-Rank | 62.57 | 91.03 | 91.32 | 77.98 | 94.27 |
| BAdam [Luo et al., 2024] | 54.44 | 89.01 | 91.35 | 68.59 | 94.15 |
| GaLore [Zhao et al., 2024] | 58.54 | 90.61 | 91.30 | 74.37 | **94.50** |
| LDAdam [Robert et al., 2025] | **58.81** | 90.90 | **92.22** | 76.53 | 94.27 |
| **SubTrack++** (Ours) | 58.55 | **90.95** | 92.04 | **78.34** | 90.02 |

Table 8: Evaluating the performance of SubTrack++ and other baselines when fine-tuning RoBERTa-Large on SuperGLUE tasks with $r = 8$. The performance is measured via Accuracy (↑) for COPA, WIC, WSC, BoolQ, and $AX_g$ tasks, and F1 (↑) for CB. The best results are marked in **bold**, with the second-best performance underlined.

|  | BoolQ | CB | COPA | WIC | WSC | $AX_g$ |
|---|---|---|---|---|---|---|
| Full-Rank | 85.96 | 90.33 | 76.00 | 71.79 | 63.46 | 96.30 |
| GaLore [Zhao et al., 2024] | 85.44 | **88.85** | 80.00 | **71.47** | 63.46 | **100.00** |
| BAdam [Luo et al., 2024] | 82.51 | 53.28 | 59.00 | 70.38 | 60.58 | 51.85 |
| LDAdam Robert et al. [2025] | **85.75** | 56.16 | 80.00 | 71.00 | **64.42** | 70.37 |
| **SubTrack++** (Ours) | 85.38 | 83.96 | **82.00** | 70.70 | 62.5 | **100.00** |

Table 9: Comparing final evaluation loss (↓), wall-time, and memory of fine-tuning Llama-2-7B-chat-hf on Alpaca dataset. cd

| Method | Evaluation Loss | Wall-Time (min) | Memory (GB) |
|---|---|---|---|
| GaLore [Zhao et al., 2024] | 0.88 | 178 | 59.4 |
| LDAdam [Robert et al., 2025] | 0.85 | 342 | 66.1 |
| SubTrack++ (Ours) | 0.88 | 117 | 62.5 |

Table 10: Hyperparameters of fine-tuning RoBERTa-Base on GLUE tasks.

|  |  | SST-2 | MRPC | CoLA | RTE | STS-B |
|---|---|---|---|---|---|---|
| Shared Parameters | Batch Size | 16 | 16 | 32 | 16 | 16 |
|  | # Epochs |  |  | 30 |  |  |
|  | Max Seq. Len. |  |  | 512 |  |  |
| Low-Rank Optimizer Methods Parameters | Learning Rate | 2E-05 | 2E-05 | 1E-05 | 2E-05 | 3E-05 |
|  | SubTrack Step-Size | 0.1 | 3.0 | 5.0 | 15.0 | 10.0 |
|  | Subspace Update Interval |  |  | 500 |  |  |
|  | Rank Config |  |  | 8 |  |  |
|  | $\alpha$ |  |  | 2 |  |  |
| BAdam Parameters | Learning Rate | 2E-05 | 2E-05 | 1E-05 | 2E-05 | 3E-05 |
|  | Block Switch Interval |  |  | 100 |  |  |
|  | Switch Mode |  |  | Random |  |  |

Table 11: Hyperparameters of fine-tuning RoBERTa-Large on SuperGLUE tasks.

|  |  | BoolQ | CB | COPA | WIC | WSC | $AX_g$ |
|---|---|---|---|---|---|---|---|
| Shared Parameters | Batch Size |  |  | 16 |  |  |  |
|  | # Epochs |  |  | 30 |  |  |  |
|  | Learning Rate |  |  | 1e-5 |  |  |  |
|  | Max Seq. Len. |  |  | 512 |  |  |  |
| Low-Rank Optimizer Methods Parameters | SubTrack++ Step-Size | 0.1 | 10.0 | 10.0 | 100.0 | 1.0 | 1.0 |
|  | Subspace Update Interval | 500 | 100 | 100 | 500 | 250 | 100 |
|  | Rank Config. |  |  | 8 |  |  |  |
|  | $\alpha$ |  |  | 4 |  |  |  |
| BAdam Parameters | Block Switch Interval | 100 | 50 | 50 | 100 | 50 | 50 |
|  | Switch Mode |  |  | Random |  |  |  |

Table 12: Hyperparameters of fine-tuning Llama-2-7B-chat-hf on Alpaca dataset.

| Parameter | Value |
|---|---|
| Subspace Update Interval | 500 |
| Rank | 1024 |
| $\alpha$ | 0.25 |
| Target Modules | att, mlp |
| Batch Size | 8 |
| Epoch | 1 |

