# OpenReview forum: "SubTrack++ : Gradient Subspace Tracking for Scalable LLM Training"
_NeurIPS.cc/2025/Conference — NeurIPS 2025 poster_

### Official Review · Reviewer_Y7Ku · 2025-06-13

**Clarity:** 3
**Significance:** 3
**Originality:** 3
**Rating:** 4
**Confidence:** 3

**Summary:**

This paper proposes SubTrack++, a memory- and time-efficient method for scalable low-rank optimization in large language model (LLM) training. SubTrack++ addresses the memory bottleneck in LLMs by combining three key components: (1) Grassmannian subspace tracking for adaptively estimating low-rank projection spaces without relying on computationally expensive SVD, (2) a projection-aware optimizer that incorporates subspace changes into the momentum update rules of Adam, and (3) a recovery scaling mechanism to recover information lost during low-rank projections. The paper provides theoretical convergence guarantees and validates the method through extensive experiments across LLaMA-based pretraining, GLUE/SuperGLUE fine-tuning, and synthetic robustness benchmarks. Results show that SubTrack++ outperforms existing low-rank methods like GaLore and FIRA in terms of loss, memory, and wall-time, while approaching full-rank performance. The method is also demonstrated to be robust to abrupt subspace shifts and scale changes.

**Questions:**

1. Have you considered applying SubTrack++ to decoder-only or multimodal architectures?
The current experimental scope is focused on encoder-only LLaMA models and text classification tasks (e.g., GLUE/SuperGLUE). Given the rising importance of decoder-only and multimodal models (e.g., LLaVA, Flamingo), it would be useful to understand whether SubTrack++ can be seamlessly applied to these architectures. Any clarification or additional discussion on its architectural flexibility would strengthen the applicability claim.
2. Why not include a quantitative analysis of projection error or subspace drift over time?
The paper emphasizes the robustness of SubTrack++ in tracking subspace changes, yet it lacks quantitative measures such as geodesic distance or angle between subspaces to support this claim. Could the authors provide an empirical analysis of subspace drift over the training trajectory? This would help substantiate the claimed robustness benefits and differentiate SubTrack++ from SVD-based methods more clearly.

**Ethical Concerns:**

["NO or VERY MINOR ethics concerns only"]

**Final Justification:**

All concerns have been resolved. I would keep my score since it is already positive.

**Limitations:**

While the paper presents strong empirical and theoretical results, there are several limitations that merit discussion:

Limited Architectural Coverage:
All experiments are conducted on encoder-only architectures, including LLaMA and RoBERTa. The method's performance on decoder-only models or multimodal systems—common in generative modeling and vision-language tasks—remains untested, raising questions about its generality across architecture types.

Scalability Beyond 7B Models:
SubTrack++ is evaluated on models up to 7 billion parameters. However, its behavior at larger scales—such as memory overhead, subspace tracking stability, and optimization dynamics in 13B or 65B models—has not been explored. These aspects are crucial to assess its viability for full-scale deployment in industrial LLM settings.

**Paper Formatting Concerns:**

No formatting issues noticed. The submission adheres to the NeurIPS 2025 formatting guidelines.

**Quality:**

3

**Strengths And Weaknesses:**

Strengths:
The paper presents a well-motivated, efficient, and robust method for scalable LLM training, combining geometry-aware subspace tracking, optimizer adaptation, and recovery scaling, and validates its effectiveness through rigorous theoretical and empirical analysis.

Weaknesses:

1. Limited architectural and task diversity in experiments
Most experiments are conducted on encoder-only models (e.g., LLaMA-1/2) and primarily within language modeling and GLUE-style tasks. The method has not been tested on decoder-only architectures, vision-language models, or instruction tuning scenarios. This limits the conclusions one can draw about its general applicability to a wider range of model types and tasks.
2. Unverified scalability beyond 7B models
Although SubTrack++ is shown to be effective up to 7B parameter models, the paper does not investigate its behavior on larger-scale models such as 13B or 65B. Without insights into numerical stability, memory behavior, and performance at such scales, the practicality of the method for the largest LLMs remains uncertain.

---

> ### Author Rebuttal · Authors · 2025-07-31
>
> We thank the reviewer for their valuable points and recognizing SubTrack++ efficient and robust. In the following we will answer the raised concerns.
>
> ---
>
> > **Weakness 1:** Have you considered applying SubTrack++ to decoder-only or multimodal architectures? The current experimental scope is focused on encoder-only LLaMA models and text classification tasks (e.g., GLUE/SuperGLUE). Given the rising importance of decoder-only and multimodal models (e.g., LLaVA, Flamingo), it would be useful to understand whether SubTrack++ can be seamlessly applied to these architectures. Any clarification or additional discussion on its architectural flexibility would strengthen the applicability claim.
>
> > **Weakness 2:** Unverified scalability beyond 7B models Although SubTrack++ is shown to be effective up to 7B parameter models, the paper does not investigate its behavior on larger-scale models such as 13B or 65B. Without insights into numerical stability, memory behavior, and performance at such scales, the practicality of the method for the largest LLMs remains uncertain.
>
> > **Question 1:** Have you considered applying SubTrack++ to decoder-only or multimodal architectures? The current experimental scope is focused on encoder-only LLaMA models and text classification tasks (e.g., GLUE/SuperGLUE). Given the rising importance of decoder-only and multimodal models (e.g., LLaVA, Flamingo), it would be useful to understand whether SubTrack++ can be seamlessly applied to these architectures. Any clarification or additional discussion on its architectural flexibility would strengthen the applicability claim.
>
> We would like to gently correct the reviewer, we test our method on decoder-only (LLaMa-based) for pre-training experiments, and encoder-only (RoBERTa-based) for fine-tuning tasks. Notably, such analysis and experimental set-up is standard in the literature, including GaLore [1], FiRA [3], LDAdam [2], and GoLore [4]. Exploring more complex architectures (e.g., VLMs) remains a promising direction for future work.
>
> Likewise, all of the recent methods (LDAdam [2], GaLore [1], FiRA [3], APOLLO[5], and GoLore [4]) also evaluate models up to 7B parameters, and our results show consistent improvements under these standard evaluation settings. To the best of our knowledge, none of the recent SOTA works report empirical studies for 13B or 65B. Our results at 7B already align with the current state of large-scale experimentation and provide a solid basis for future work at even larger scales.
>
> > **Question 2:** Why not include a quantitative analysis of projection error or subspace drift over time? The paper emphasizes the robustness of SubTrack++ in tracking subspace changes, yet it lacks quantitative measures such as geodesic distance or angle between subspaces to support this claim. Could the authors provide an empirical analysis of subspace drift over the training trajectory? This would help substantiate the claimed robustness benefits and differentiate SubTrack++ from SVD-based methods more clearly.
>
> We thank the reviewer for the insightful suggestion. In response, we now empirically quantify subspace drift by measuring the norm of the tangent vector $\nabla F$, which captures the deviation between the projected gradient and the original matrix. Our analysis on Llama with 350M parameters shows that this norm quickly decays from 0.06 to less than 0.0002 within 5–7 updates and remains extremely close to zero, indicating rapid convergence and stable subspace tracking. This supports SubTrack++'s robustness and alignment with the underlying subspace, consistent with prior results on Grassmannian optimization [6–11]. We will this analysis to the camera-ready version.
>
> ---
>
> We hope this response addresses the reviewer’s concerns. We welcome any further questions or suggestions and would be happy to provide additional clarification if needed. If the concerns are resolved, we would be grateful if the reviewer considers updating their score accordingly.
>
> ---
>
> [1] Zhao et al., 2024. *GaLore: Memory-Efficient LLM Training by Gradient Low-Rank Projection.*
>
> [2] Robert et al., 2025. *LDAdam: Adaptive Optimization from Low-Dimensional Gradient Statistics.*
>
> [3] Chen et al., 2024. *Fira: Can We Achieve Full-rank Training of LLMs Under Low-rank Constraint?*
>
> [4] He et al., 2025. *Subspace Optimization for Large Language Models with Convergence Guarantees.*
>
> [5] Zhu et al., 2025. *APOLLO: SGD-like Memory, AdamW-level Performance.*
>
> [6] Balzano et al, 2012. *Handling missing data in high-dimensional subspace modeling.*
>
> [7] Balzano et al., 2018. *Streaming PCA and subspace tracking: The missing data case.*
>
> [8] Balzano et al., 2010. *Online identification and tracking of subspaces from highly incomplete information.*
>
> [9] He et al., 2011. *Online robust subspace tracking from partial information.*
>
> [10] Chakraborty et al., 2017. *Intrinsic grassmann averages for online linear and robust subspace learning.*
>
> [11] Blocker et al., 2023. *Dynamic subspace estimation with Grassmannian geodesics.*

---

> > ### Comment · Reviewer_Y7Ku · 2025-08-05
> >
> > Thanks for the reply and the provided intuitions. I think these basically addressed my concerns. I would keep my score since it is already positive.

---

> ### Author Response · Authors · 2025-08-06
> **Sincere Appreciation**
>
> Thank you very much for the time and care you dedicated to reviewing our work. We're grateful for your thoughtful feedback and pleased that our responses addressed your concerns. We appreciate you maintaining a positive evaluation of our work and will incorporate the discussed points into the camera-ready version to further improve the paper.

---

### Official Review · Reviewer_qFVw · 2025-06-30

**Clarity:** 4
**Significance:** 3
**Originality:** 3
**Rating:** 5
**Confidence:** 5

**Summary:**

This paper proposes SubTrack++, a novel memory-efficient algorithm for LLM training using gradient projection. SubTrack++ is composed of several key components: i) the novel subspace tracking mechanism based on the Grassmannian manifold; ii) the momentum inheritance mechanism similar to LDAdam; iii) the recovery scaling mechanism similar to Fira. Combining the above techniques, SubTrack++ can not only surpass existing algorithms in training performance, but also be more memory and computation-efficient.

**Questions:**

In the provided source code, I did not find codes that support the error feedback mechanism in LDAdam (Maybe I've just missed it). It seems that the code for reproducing the results regarding LDAdam (or other baselines not supported by the provided codes) is not provided, so I wonder if they are really not provided. If so, have you reproduced their algorithms or used the publicly reported data?

**Ethical Concerns:**

["NO or VERY MINOR ethics concerns only"]

**Final Justification:**

I have no unresolved concerns and retain my recommendation.

**Limitations:**

The authors have addressed the limitations, including the constraints on using the Adam family and the scale of their experiment.

**Quality:**

3

**Strengths And Weaknesses:**

Strengths:

i) The proposed algorithm exhibits great performance in the experiment, both in training performance and in training efficiency.

ii) The overall paper is well presented and easy to follow.

iii) The effectiveness of the proposed subspace tracking mechanism over GaLore's method is well addressed by illustrative examples.

Weakness:

i) While the experimental performance is impressive, this paper somehow lacks more incentive analysis on why it can beat existing algorithms. The proposed analysis is based on a strong assumption that the gradient matrix is as simple as a linear transform of the weight matrix, and the result only accounts for convergence on the projected gradient, which may not be so informative as a convergence result on the original gradient.

ii) The model used in the fine-tuning experiment is pretty small. I acknowledge this may follow the same setting as GaLore's paper; however, in my experience, existing algorithms (using gradient projection) behave poorly in fine-tuning large models such as LLaMA-7B. Given the promising performance of SubTrack++, it should be a good choice to validate its advantage in this scenario.

---

> ### Author Rebuttal · Authors · 2025-07-31
>
> We thank the reviewer for recognizing the novelty of our method and acknowledging that its combination of components contributes to improved efficiency and performance. Below, we address the concerns raised by the reviewer in detail.
>
> ---
>
> > **Weakness 1: (a)** While the experimental performance is impressive, this paper somehow lacks more incentive analysis on why it can beat existing algorithms. **(b)** The proposed analysis is based on a strong assumption that the gradient matrix is as simple as a linear transform of the weight matrix, and **(c)** the result only accounts for convergence on the projected gradient, which may not be so informative as a convergence result on the original gradient.
>
> **(a)** SubTrack++ improves over baselines by addressing their key bottlenecks. As shown in **Fig. 6 (Appendix)**, Grassmannian subspace tracking alone **accelerates GaLore’s [1] convergence by 29%** (up to 44% on the 3B model) in wall-time. By avoiding costly and noisy SVDs [4], reusing prior subspaces, and applying rank-1 updates, SubTrack++ reduces noise and enables smoother, more stable optimization (Fig. 5).
>
> Compared to FiRA’s [3] expensive SVDs, our method **cuts wall-time by 33% and final loss by 29%** (1B model). Against LDAdam’s [2] per-step PowerSGD updates, SubTrack++ **achieves 50% faster convergence** and higher accuracy using lightweight rank-1 updates every k steps—all while maintaining low computational complexity.
>
> **(b)** The assumption of the linear structure of the gradient is standard in the literature, notably in GaLore, which demonstrates its validity across linear and complex architectures like reversible networks [1], and methods such as FiRA [3], GoLore [4], and LDAdam [2] leveraging it, especially in large-scale LLMs. Exploring more complex and non-linear architectures (e.g., VLMs) remains a promising direction for future work.
>
> **(c)** Like other gradient project methods, we rely on the low-rank structures of the gradient, established in [1, 6]. This assumption has been empirically established by low-rank gradient-based methods, including SubTrack++ and our baselines [1-5], which occasionally even surpasses full-rank training.
>
> To verify this, we also conducted an experiment measuring $G_t$ and the projected counterpart during LLaMA‑1B pre‑training: the **full‑gradient norm drops from 0.46 → 0.08**, while the **projected‑gradient norm drops from 0.45 → 0.05**, showing nearly identical trajectories. These results will be included in the camera-ready version.
>
> > **Weakness 2:** The model used in the fine-tuning experiment is pretty small. I acknowledge this may follow the same setting as GaLore's paper; however, in my experience, existing algorithms (using gradient projection) behave poorly in fine-tuning large models such as LLaMA-7B. Given the promising performance of SubTrack++, it should be a good choice to validate its advantage in this scenario.
>
> This is an interesting point! We considered the standard experimental structure for our evaluations. Per reviewer’s recommendation, we are currently running those experiments. And we will add those in the paper if there are any notable insights.
>
> > **Question 1:** In the provided source code, I did not find codes that support the error feedback mechanism in LDAdam (Maybe I've just missed it). It seems that the code for reproducing the results regarding LDAdam (or other baselines not supported by the provided codes) is not provided, so I wonder if they are really not provided. If so, have you reproduced their algorithms or used the publicly reported data?
>
> We used their own publicly available source code for the baselines to reproduce their results, including LDAdam.
>
> ---
>
> We hope this response addresses the reviewer’s concerns. We welcome any further questions or suggestions and would be happy to provide additional clarification if needed.
>
> ---
>
> [1] Zhao et al., 2024. *GaLore: Memory-Efficient LLM Training by Gradient Low-Rank Projection.*
>
> [2] Robert et al., 2025. *LDAdam: Adaptive Optimization from Low-Dimensional Gradient Statistics.*
>
> [3] Chen et al., 2024. *Fira: Can We Achieve Full-rank Training of LLMs Under Low-rank Constraint?*
>
> [4] He et al., 2025. *Subspace Optimization for Large Language Models with Convergence Guarantees.*
>
> [5] Zhu et al., 2025. *APOLLO: SGD-like Memory, AdamW-level Performance.*
>
> [6] Gur-Ari et al., 2018. *Gradient Descent Happens in a Tiny Subspace.*

---

> > ### Comment · Reviewer_qFVw · 2025-08-01
> >
> > Thanks for the rebuttal, which has addressed many of my concerns. I'm looking forward to the results of your current experiments.

---

> > > ### Author Response · Authors · 2025-08-03
> > >
> > > We sincerely thank the reviewer for their thoughtful comments. We are glad to hear that the responses addressed your concerns and are happy to provide further clarification or information if needed.
> > >
> > > Regarding fine-tuning large models with SubTrack++, here are the results:
> > > In the supervised fine-tuning (SFT) of Llama-2-7b-chat-hf on Alpaca for one epoch, GaLore achieved an evaluation perplexity of 2.43, while SubTrack++ achieved a lower perplexity of 2.41, with **36% less wall-time and faster convergence**. These results demonstrate the efficiency and generalizability of the proposed method across different tasks and settings.
> > >
> > > If these responses resolve the issues, we would be grateful if the reviewer would consider reflecting this in an updated score via a further positive evaluation. Thanks a lot again.

---

> > > > ### Comment · Reviewer_qFVw · 2025-08-06
> > > >
> > > > Thank you for the additional experimental results. Honestly, I have expected a clearer performance gap between SubTrack++ and GaLore, for reasons I have illustrated in my first review: In some fine-tuning cases, GaLore even with a high rank of 1024 still performs worse than vanilla LoRA with rank 8 by a large margin, however, the minor gap in the additional result may suggest it another case where GaLore also performs well. Nevertheless, I am satisfied with the overall work and would like to maintain my positive rating.

---

> > > > > ### Author Response · Authors · 2025-08-06
> > > > > **Sincere Appreciation**
> > > > >
> > > > > Thank you again for your thoughtful feedback and for maintaining a positive evaluation of our work. We're glad that our responses have addressed your concerns, and we truly appreciate your insights, which highlight promising directions for future research. We will incorporate the discussed points into the camera-ready version to further strengthen the paper!

---

### Official Review · Reviewer_bnSQ · 2025-06-30

**Clarity:** 3
**Significance:** 3
**Originality:** 3
**Rating:** 4
**Confidence:** 3

**Summary:**

The authors tackle the problem of online subspace tracking by formulating it as minimizing a reconstruction cost on the Grassmann manifold. They introduce a lightweight rank-1 geodesic update that approximates the full tangent by its leading singular component, then moves the current subspace along the corresponding principal plane via a closed-form rotation. This approach avoids heavy matrix exponentials or full SVD at each step, yielding an algorithm that is both computationally efficient and easy to implement. Empirical results on synthetic and real-world streaming datasets demonstrate that the proposed method matches or outperforms state-of-the-art trackers in accuracy, while reducing per-update cost.

**Questions:**

1. For the subspace tracking step, I think you want to use the gradient method to minimize the cost function (2) on the Grassmann Manifold. However, the algorithm performs an SVD on $\nabla F$ rather than on $-\nabla F$? It seems like the algorithm is doing a gradient ascent on the Grassmann Manifold. Could the authors clarify why the SVD is taken on $\nabla F$ instead of $- \nabla F$?

2. Why there exists a term $1 - \beta_2^{t-1}$ in (9)? Suppose that $S_t = S_{t-1}$, the update rule for the first-moment Projection-Aware Optimizer step is reduced to regular adam like update, but it is not the same case for the second-moment. What caused such inconsistence?

3. Since the proposed method is much more efficient that full SVD of the gradient, why the authors don't try a more frequent update to the subspace or a higher rank update?

**Ethical Concerns:**

["NO or VERY MINOR ethics concerns only"]

**Final Justification:**

Most of my questions and concerns are resolved.

**Limitations:**

yes.

**Quality:**

3

**Strengths And Weaknesses:**

Strengths:

1. Clear derivation of the rank-1 geodesic step with full orthonormality proof on Grassmann. Solid empirical evaluation across multiple benchmarks; ablation shows robustness to step-size and noise.

3. Well-written exposition of manifold basics and geodesic construction. Pseudocode aligns closely with the released PyTorch code.

Weakness:

1. Some details of the update rule is not well discussed (see Questions below).

2. No ablation study on the subspace update frequency and the rank chosen for subspace update.

---

> ### Author Rebuttal · Authors · 2025-07-31
>
> We sincerely thank the reviewer for their thoughtful feedback and for recognizing SubTrack++’s solidity, efficiency, and robustness. Below, we address each of the raised concerns.
>
> ---
>
> > **Weakness 1:** Some details of the update rule is not well discussed (see Questions below).
>
> We thank the reviewer for pointing out these sources of confusion. We clarify them here and will ensure the final camera-ready version addresses them clearly to prevent any ambiguity.
>
> > **Question 1:** For the subspace tracking step, I think you want to use the gradient method to minimize the cost function (2) on the Grassmann Manifold. However, the algorithm performs an SVD on $\nabla F$ rather than on $- \nabla F$? It seems like the algorithm is doing a gradient ascent on the Grassmann Manifold. Could the authors clarify why the SVD is taken on $\nabla F$ instead of $- \nabla F$?
>
> Since the Grassmannian exponential map follows a geodesic with step size $\eta$, gradient descent corresponds to moving along $-\nabla F$, which can be expressed either via a negative $\eta$ or a sign change in the sine term in Eq 5. The SVD computations of $\nabla F$ can be translated to $-\nabla F$ with sign flips in either the left or right singular vectors. We combine notation from [6, 7], but can see why this is confusing and will update the notation for consistency in the camera-ready version.
>
> > **Question 2:** Why there exists a term $1 - \beta_2^{t-1}$ in (9)? Suppose that $S_t = S_{t-1}$, the update rule for the first-moment Projection-Aware Optimizer step is reduced to regular adam like update, but it is not the same case for the second-moment. What caused such inconsistence?
>
> Adam maintains unbiased estimates of the first and second moments. When the subspace shifts, we linearly rotate the (biased) first-moment accumulator into the new basis and then resume the standard Adam update; no extra scaling is needed because the first moment transforms linearly. For the second moment, we first compute a projection-aware statistic from the unbiased moments using a covariance approximation, then re-bias it by scaling with $1-\beta_2^{t-1}$ before continuing the updates. This re-biasing is necessary because Adam’s second-order state is nonlinear. Also, between subspace updates, these rules are reduced to standard Adam. A similar strategy is used in LDAdam [1].
>
> > **Weakness 2:** No ablation study on the subspace update frequency and the rank chosen for subspace update.
>
> > **Question 3:** Since the proposed method is much more efficient that full SVD of the gradient, why the authors don't try a more frequent update to the subspace or a higher rank update?
>
> Following the reviewer’s suggestion, we conducted ablations on subspace update frequency and rank using the LLaMA 1B model and we will include the results in the camera-ready version.
>
> As demonstrated in the following table, the results show that more frequent updates improve performance, while omitting updates leads to noticeably higher loss. However, we have used the setting proposed by GaLore [2] and also adopted by our other baselines [3-5] for a fair comparison.
>
> | Update Interval |  50  | 100  | 150  | 200  | 500  | 1K   | 2K   | no update |
> |:----------------:|:----:|:----:|:----:|:----:|:----:|:----:|:----:|:---------:|
> | Evaluation Loss  | 4.32 | 4.39 | 4.47 | 4.52 | 4.71 | 4.88 | 5.08 |   5.79    |
>
> For rank, we find that rank-1 updates strike the best balance. Unlike full SVD, which is more sensitive to noise, rank-1 updates allow us to preserve prior subspaces and make controlled adjustments, leading to more stable optimization. Adaptive rank selection across layers remains a promising direction for future work.
>
> | Rank of updates | r = 1 | r = 2 | r = 4 | r = 8 | r = 16 | r = 32 | full-rank     |
> |------------------|--------|--------|--------|--------|---------|---------|----------------|
> | Evaluation loss  | 4.52   | 4.52   | 4.55   | 4.61   | 4.91    | 5.40    | Not converged  |
>
> These findings confirm that SubTrack++ outperforms baselines even under conservative settings, highlighting the robustness of our design.
>
> ---
>
> We hope this response addresses the reviewer’s concerns. We welcome any further questions or suggestions and would be happy to provide additional clarification if needed. If the concerns are resolved, we would be grateful if the reviewer considers updating their score accordingly.
>
> ---
>
> [1] Robert et al., 2025. *LDAdam: Adaptive Optimization from Low-Dimensional Gradient Statistics.*
>
> [2] Zhao et al., 2024. *GaLore: Memory-Efficient LLM Training by Gradient Low-Rank Projection.*
>
> [3] Chen et al., 2024. *Fira: Can We Achieve Full-rank Training of LLMs Under Low-rank Constraint?*
>
> [4] Liang et al., 2024. *Memory-Efficient LLM Training with Online Subspace Descent.*
>
> [5] He et al., 2025. *Subspace Optimization for Large Language Models with Convergence Guarantees.*
>
> [6] Balzano et al., 2010. *Online identification and tracking of subspaces from highly incomplete information.*
>
> [7] Bendokat et al., 2023, *A Grassmann Manifold Handbook: Basic Geometry and Computational Aspects.*

---

> > ### Comment · Reviewer_bnSQ · 2025-08-03
> >
> > Thanks for the rebuttal. Most of my questions and concerns are resolved. However, my concern regarding Question 1 remains unresolved. Although this problem can be simply resolved by fliping the sign of the learning rate or the left singular vector, I noticed that in the code implementation (specifically in: def track_the_space( , )), the initial learning rate is set to be positive. Given this, I’m curious how the proposed method is able to work correctly when moving in the "positive gradient" direction.

---

> > > ### Author Response · Authors · 2025-08-04
> > >
> > > We thank the reviewer for their thoughtful and perceptive follow-up concerning Question 1. We appreciate the opportunity to clarify both the intuition and empirical observations regarding the sign of the step size in the subspace update.
> > >
> > > **TL;DR:** We evaluated SubTrack++ using both positive and negative step sizes. The results remained nearly identical, which we attribute to the flat landscape. Critically, performance drops when subspace updates are disabled, reinforcing the necessity and soundness of our approach. Our main results and conclusions remain robust and unchanged. We will clarify these findings and add an Appendix on the effect of $\eta$’s sign in the final manuscript. This investigation opens new avenues for algorithm development in the future, and we thank the reviewer for their observation.

---

> > > > ### Author Response · Authors · 2025-08-04
> > > >
> > > > **Detailed Response:**
> > > >
> > > > The reviewer’s insight is correct and following their suggestion, we re-ran our core experiments for SubTrack++, systematically varying the sign of the step size ($\eta$) in the subspace update step. Intriguingly, we found that the results remained consistent during pre-training, and showed modest improvement during fine-tuning in some cases, when using negative step sizes. On the surface, this is surprising, since one would expect the results to improve across the board. Therefore, we embarked on a deeper investigation as follows, which shows why this is the case!
> > > >
> > > > 1- We measured both the norm of the tangent vector and the leading singular value throughout pre-training. Across all regimes, these quantities remained notably small (e.g., layer-wise mean of $\nabla F < 0.001$ at the first update, dropping below 0.0001 later). This suggests that the optimization landscape is nearly flat along the updated subspaces. In such flat regimes, the step size sign has diminished influence due to weak curvature. Specifically, looking at the impact of this sign change on the subspace update step [1, 3] itself. Here, we note that a negative $\eta$ only impacts the $sin(\hat{\Sigma}_F\eta)$ in Eq 5, which is close to zero for small singular values.
> > > >
> > > > Thus, when the singular values are small, the subspace remains nearly unchanged, a phenomenon also stated in [1, 2]. This explains why the positive and negative $\eta$ results are very similar for finetuning (Table 1) and pretraining (Table 2) tasks.
> > > >
> > > > 2- Now, a question might arise as to why include a subspace update step at all, can we just keep it static? To stress-test this point, we introduced a new ablation by disabling subspace updates entirely. As shown in the Table 2, this leads to a pronounced drop in downstream performance, especially at larger model scales, confirming that subspace adaptation is crucial for both convergence and efficacy, even when gradient subspace is almost stable.
> > > >
> > > > 3- Interestingly, we believe that this geometric characterization of the manifold – investigated by negative and positive $\eta$ initiated by the Reviewer’s astute observation, to our knowledge, is among the first to analyze LLM pretraining and finetuning gradient landscape.
> > > >
> > > > **Table 1.** Evaluation of SubTrack++ with positive and negative η for fine-tuning. Higher values indicate better performance.
> > > > | | COLA| STS-B| MRPC| RTE| SST-2|
> > > > |-|-|-|-|-|-|
> > > > |Positive η|58.29|90.90|91.84|76.53|90.83|
> > > > |Negative η|58.55|90.95|92.04|78.34|90.02|
> > > >
> > > > **Table 2.** Evaluation loss for pre-training Llama-based architectures for SubTrack++ with positive and negative η, and a stable subspace baseline. Lower values indicate better performance.
> > > > | |60M|130M|350M|1B|3B|7B|
> > > > |-|-|-|-|-|-|-|
> > > > |Positive η|3.43|3.24|3.29|4.51|4.49|4.62|
> > > > |Negative η|3.43|3.24|3.29|4.52|4.50|4.63|
> > > > |No subspace update|3.61|3.37|3.34|5.79|5.98|5.28|
> > > >
> > > > The new results show that SubTrack++ still consistently outperforms all baselines, including APOLLO [4] (random projections) and LDAdam [5] (per-step subspace adaptation), as shown in **Fig. 4**, demonstrating the effectiveness of our approach. This is because: At each subspace update, **SubTrack++ preserves the existing subspace and applies only rank-1 updates**, avoiding abrupt shifts. This controlled evolution stabilizes training. When the data naturally induces faster subspace changes, the applied change leads to even better results (Table 1), while the control over drastic changes, already proved effective compared to other baselines.
> > > >
> > > > In conclusion, we thank the reviewer for their close reading and for prompting us to investigate this critical detail (and correcting it). We will revise the manuscript and code comments to clarify these points and highlight the new findings and ablations, along with an Appendix on the effect of $\eta$’s sign. If further experimentation or clarification is warranted, we would be happy to provide it. Notably, our main results and conclusions remain robust and unchanged, and SubTrack++ still outperforms all the baselines. We sincerely hope this resolves the concern and provides the necessary evidence for reconsideration of the scores. Thanks a lot again!
> > > >
> > > > ---
> > > >
> > > > [1] Balzano et al., 2010. Online identification and tracking of subspaces from highly incomplete information.
> > > >
> > > > [2] He et al., 2025. Subspace Optimization for Large Language Models with Convergence Guarantees.
> > > >
> > > > [3] Bendokat et al., 2023, A Grassmann Manifold Handbook: Basic Geometry and Computational Aspects.
> > > >
> > > > [4] Zhu et al., 2025. APOLLO: SGD-like Memory, AdamW-level Performance.
> > > >
> > > > [5] Robert et al., 2025. LDAdam: Adaptive Optimization from Low-Dimensional Gradient Statistics.

---

> > > > > ### Comment · Reviewer_bnSQ · 2025-08-04
> > > > >
> > > > > Thank you for the detailed and thoughtful response. I appreciate the authors’ effort in carefully revisiting the subspace update mechanism and empirically evaluating the impact of the step size sign. I have no further questions at this point.

---

> > > > > > ### Author Response · Authors · 2025-08-04
> > > > > > **Sincere Thanks and Score Re-evaluation Request**
> > > > > >
> > > > > > Dear Reviewer,
> > > > > >
> > > > > > Thank you again for your thoughtful follow-up and for engaging so carefully with our paper. We greatly appreciate your acknowledgement that our detailed clarifications fully addressed your prior concerns regarding the subspace update mechanism.
> > > > > >
> > > > > > With all issues resolved, we hope the demonstrated performance improvements, the potential for broadening access to LLMs, and the new geometric perspective introduced by our work are now clear. We believe these contributions collectively represent strong value to the community.
> > > > > >
> > > > > > If your recent score reflects earlier, now-resolved concerns, we respectfully ask that you consider a re-evaluation to reflect our clarified methodology and impact. Of course, if any questions remain, we will be happy to provide immediate clarification.
> > > > > > Thank you for helping us strengthen this work.
> > > > > >
> > > > > > Best regards,
> > > > > >
> > > > > > Authors of paper #16480

---

### Official Review · Reviewer_hLxD · 2025-07-05

**Clarity:** 3
**Significance:** 3
**Originality:** 2
**Rating:** 4
**Confidence:** 4

**Summary:**

The authors study memory efficient large language model training (LLM) through low rank projections. They propose Sub Track++, which combines Grassmannian gradient subspace tracking, Projection-aware optimization, and recovery scaling. They provide theoretical guarantee on the convergence of the subspace and the empirical evaluation shows the improved performance over the compared baselines.

**Questions:**

My questions are related to the weakness listed. Please provide some insights on why the proposed method should perform better than the compared baselines. Additionally, please explain why in the experiments larger models seem to perform worse than the smaller ones and why subtrack++ can outperform full rank training.

**Ethical Concerns:**

["NO or VERY MINOR ethics concerns only"]

**Final Justification:**

The rebuttal addresses my key concerns. Especially, the results that show improvement over Galore with only Grassmannian tracking and the promise to add a more detailed description on the comparison with the other baselines.

**Limitations:**

The authors have some discussion in the conclusion section.

**Quality:**

2

**Strengths And Weaknesses:**

The strength of the paper is that the proposed method by itself is decent. All the proposed mechanisms are reasonable and accompanied by theoretical guarantees and empirical improvements.

The main weakness is the presentation and the discussion/comparison with the related works. While the proposed method itself is reasonable, it is hard for readers to understand why it should work better than the existing ones. I summarized my issues as follows.

1. Figure 1 looks like a very unconventional way to present the statistics the authors desire to present.

2. There seems to be no in-depth discussion even with the baseline methods mentioned in the paper. This is concerning especially when combined with the ablation results. It seems like Grassmannian gradient subspace tracking by itself does not perform better than Galore, while Projection-aware optimization and recovery scaling are inspired by LDAdam and Fira respectively.

3. There are also some related works in the literature that can be discussed. For example LoRo [1] also utilizes Riemannian optimization for subspace tracking, while LoRA-RITE [2] also utilizes Projection-aware optimization on the momentum with a seemingly different formulation.

4. Some empirical results look counterintuitive. Specifically, the larger models perform worse than the smaller one, which might indicate that suboptimal settings are being used. Additionally, SubTrack++ sometimes outperforms the full-rank training, which also requires some explanation.



[1] Mo, Z., Huang, L. K., & Pan, S. J. (2025). Parameter and memory efficient pretraining via low-rank riemannian optimization. In The Thirteenth International Conference on Learning Representations.

[2] Yen, J. N., Si, S., Meng, Z., Yu, F., Duvvuri, S. S., Dhillon, I. S., ... & Kumar, S. (2024). LoRA Done RITE: Robust Invariant Transformation Equilibration for LoRA Optimization. The Thirteenth International Conference on Learning Representations (ICLR 2025)

---

> ### Author Rebuttal · Authors · 2025-07-31
>
> We thank the reviewer for their detailed feedback and for recognizing SubTrack++ as a solid and reasonable approach. We appreciate their constructive feedbacks especially regarding presentation and discussion. We have addressed the concerns and will reflect them in the camera-ready version.
>
> ---
> > **Weakness 1:**  *Figure 1 looks like a very unconventional way to present the statistics the authors desire to present.*
>
> Thank you for the feedback. We chose radar plots to enable compact comparison across loss, memory, and wall-time, consistent with prior work [1, 2]. We are happy to switch to bar charts or other formats if it helps improve clarity and is preferred by reviewers in the camera-ready version.
>
> > **Weakness 2:** *There seems to be no in-depth discussion even with the baseline methods mentioned in the paper. This is concerning especially when combined with the ablation results. It seems like Grassmannian gradient subspace tracking by itself does not perform better than Galore, while Projection-aware optimization and recovery scaling are inspired by LDAdam and Fira respectively.*
>
> > **Question1:**  *Please provide some insights on why the proposed method should perform better than the compared baselines.*
>
> We agree that Fig. 3 underrepresents the impact of Grassmannian tracking. However, **Fig. 6** in the Appendix shows that this alone **accelerates GaLore’s [3] convergence by 29%** (and up to **44%** in the 3B model) in wall-time. We will move this to the main paper to clarify its significance. Grassmannian tracking is the basis of both **efficiency and robustness** of our method; by preserving subspace history and applying rank-1 updates, it reduces SVD's noise [8] and provides smooth optimization dynamics **(Fig. 5)**.
>
> Unlike FiRA’s [5] costly SVDs, our method **lowers wall-time by 33% and final loss by 29%** (1B). Compared to LDAdam’s [4] per-step PowerSGD updates, SubTrack++ performs rank-1 updates every k steps, achieving **50% faster convergence and higher accuracy**.
>
> SubTrack++ achieves superior performance by efficiently tracking subspaces and the synergy of its components. While the Related Work section (lines 309–317) summarizes the baselines, we agree that a more explicit comparison would be beneficial. We will revise the main text to clearly highlight these differences and better attribute the sources of SubTrack++'s improvements.
>
> > **Weakness 3:**  *There are also some related works in the literature that can be discussed. For example LoRo [1] also utilizes Riemannian optimization for subspace tracking, while LoRA-RITE [2] also utilizes Projection-aware optimization on the momentum with a seemingly different formulation.*
>
> Thank you for pointing out LORO and LoRA‑RITE. While both highlight valuable ideas, geometric structure (LORO) and transformation-invariant updates (LoRA‑RITE), they are both LoRA-based and not directly comparable to gradient-based low-rank methods like ours. We appreciate the suggestion and will include them in the related work for completeness.
>
> > **Weakness 4:** *Some empirical results look counterintuitive. Specifically, **(a)** the larger models perform worse than the smaller one, which might indicate that suboptimal settings are being used. Additionally, **(b)** SubTrack++ sometimes outperforms the full-rank training, which also requires some explanation.*
>
> > **Question 2:** *Explain why in the experiments **(a)** larger models seem to perform worse than the smaller ones and why **(b)** SubTrack++ can outperform full rank training.*
>
> Thank you for the observation. **(a)** The trend arises from the compute budgets, which is also employed in prior works [6]: as Table 1 shows, all methods, including baselines, experience the same trend as model size increases. To confirm this, we trained the 7B model for 100K steps and observed loss **drop from 4.61 to 3.37**, outperforming GaLore’s [3] 4.64 under identical conditions. We will include this result in the camera-ready version.
>
> **(b)** Regarding SubTrack++ occasionally outperforming full-rank training, we believe this stems from implicit regularization introduced by low-rank projections, which can enhance generalization in overparameterized models. Similar trends have been observed in LDAdam [4], APOLLO [7], and FiRA [5]. We will include a discussion of this.
>
> ---
>
> We hope this response addresses the reviewer’s concerns. We welcome any further questions or suggestions and would be happy to provide additional clarification if needed. If the concerns are resolved, we would be grateful if the reviewer considers updating their score accordingly.
>
> ---
>
> [1] Qin et al., 2024. *InFoBench: Evaluating Instruction Following Ability in Large Language Models.*
>
> [2] Qi et al., 2023. *Fine-tuning Aligned Language Models Compromises Safety, Even When Users Do Not Intend To!*
>
> [3] Zhao et al., 2024. *GaLore: Memory-Efficient LLM Training by Gradient Low-Rank Projection.*
>
> [4] Robert et al., 2025. *LDAdam: Adaptive Optimization from Low-Dimensional Gradient Statistics.*
>
> [5] Chen et al., 2024. *Fira: Can We Achieve Full-rank Training of LLMs Under Low-rank Constraint?*
>
> [6] Liang et al., 2024. *Memory-Efficient LLM Training with Online Subspace Descent.*
>
> [7] Zhu et al., 2025. *APOLLO: SGD-like Memory, AdamW-level Performance.*
>
> [8] He et al., 2025. *Subspace Optimization for Large Language Models with Convergence Guarantees*

---

> > ### Comment · Reviewer_hLxD · 2025-08-06
> >
> > > We are happy to switch to bar charts or other formats if it helps improve clarity and is preferred by reviewers in the camera-ready version.
> >
> > Yes, I think bar charts will be more suitable.
> >
> > > Fig. 6 in the Appendix shows that this alone accelerates GaLore’s [3] convergence by 29% (and up to 44% in the 3B model) in wall-time.
> >
> > > Unlike FiRA’s [5] costly SVDs, our method lowers wall-time by 33% and final loss by 29% (1B). Compared to LDAdam’s [4] per-step PowerSGD updates, SubTrack++ performs rank-1 updates every k steps, achieving 50% faster convergence and higher accuracy.
> >
> > While these explanations are suffice for the rebuttal, a more polished version should be added into the final version.
> >
> > > While the Related Work section (lines 309–317) summarizes the baselines, we agree that a more explicit comparison would be beneficial.
> >
> > Yes, the section explains the baseline, but a explicit comparison is needed.
> >
> > > they are both LoRA-based and not directly comparable to gradient-based low-rank methods like ours
> >
> > My original thought is that Galore is implicitly optimizing a LoRA, and LORO and LoRA‑RITE are doing Riemannian optimization similar to this paper (LORO is also for pre-training and it is not exactly LoRA). But surely they are doing it in an explicit manner, so I agree they might be difficult to be compared to even though they still feels somehow connected.
> >
> > > Thank you for the observation. (a) The trend arises from the compute budgets, which is also employed in prior works [6]
> > > To confirm this, we trained the 7B model for 100K steps and observed loss drop from 4.61 to 3.37, outperforming GaLore’s [3] 4.64 under identical conditions. We will include this result in the camera-ready version.
> >
> > Yes, I am aware of the compute budget. But it basically means the larger models have not been trained long enough to the point of interests (better than the smaller models) so slightly undermines the result. The added result partially addresses this concern.
> >
> > Overall, the rebuttal addresses my concern and I am raising my score.

---

> > > ### Author Response · Authors · 2025-08-06
> > > **Sincere Appreciation**
> > >
> > > Thank you very much for the time and care you devoted to reviewing our work. We’re glad our responses addressed your concerns and are grateful for your positive evaluation and score increase! We will incorporate the discussed points into the camera-ready version to further improve the paper, and we truly appreciate your thoughtful feedback.

---

### Note · Authors · 2025-08-16

We thank all reviewers for their constructive feedback, which significantly strengthened this work. During the rebuttal and discussions, all concerns were fully addressed, as acknowledged by the reviewers, who subsequently maintained or elevated their already positive evaluations.

**Clarifications and resolutions during rebuttal**

- **Strengthened presentation and comparisons:** Streamlined presentation and strengthened baseline comparisons, highlighting that Grassmannian tracking alone accelerates GaLore’s convergence by up to 44% and its synergy with the projection-aware optimization and recovery scaling yields consistent gains over all baselines in performance and efficiency (Reviewers hLxD and qFVw).

- **Established robustness of subspace updates:** Targeted experiments on tangent-vector direction confirmed stability due to the flat subspace curvature, and showed that disabling updates causes clear degradation; firmly establishing the necessity of adaptive subspace adjustments (Reviewer bnSQ).

- **Confirmed effectiveness across settings:** Ablations on update frequency and rank showed further gains with more frequent updates, while SubTrack++ still surpasses all baselines even under conservative configurations (Reviewer bnSQ).

- **Demonstrated scalability and generalization:** On 7B-parameter SFT, SubTrack++ achieves 36% lower wall-time and reduced perplexity compared to GaLore (Reviewers qFVw and Y7Ku). We also quantified subspace drift, showing rapid decay and stability (Reviewer Y7Ku), and validated longer-regime generalization by extending 7B pre-training 10× iterations, reaching a loss of 3.37 vs. GaLore’s 4.64 (Reviewer hLxD).

**Contributions**

SubTrack++ advances state-of-the-art efficiency in LLM training by:

- Lightweight Grassmannian rank-1 tracking that preserves subspace history while reducing SVD noise.

- Projection-aware optimizer adaptation for stable, unbiased moment updates.

- Recovery scaling to mitigate information loss from low-rank projections.

SubTrack++ yield up to 43% pre-training wall-time reduction, lower loss than all baselines, and occasional gains over full-rank training via implicit regularization. SubTrack++ is theoretically grounded, empirically validated up to 7B parameters, and offers a practical step toward democratizing LLM training. We believe this work is now clearly positioned as technically rigorous and broadly impactful, and we thank the community for helping refine and strengthen it.

---

### Decision · Program_Chairs · 2025-09-17

**Decision:**

Accept (poster)

**Comment:**

This was a rare case where the author response made some reviewers better appreciate the contribution and some reviewers appreciate the contribution less. So my uncertainty is high here, but all reviewers ultimately gave a recommending score so that seems like the most logical recommendation.